# Oleuropein-based olive leaf extract enhances muscle mitochondrial bioenergetics response to moderate – but not maximal – intensity exercise in humans

Clément Lanfranchi[1], Alba Moreno-Asso[2], Astrid M. H. Horstman[2] (ID), Sara Mistro[1],
Eugenia Migliavacca[2] (ID), Ornella Cominetti[3] (ID), Jens Stolte[3], Sylviane Métairon[3] (ID), Aurélie Hermant[2],
Ane Laura Pedersen[3] (ID), Loïc Dayon[3] (ID), Umberto De Marchi[2], Jérôme N. Feige[2] (ID), Nadège Zanou[1]
and Nicolas Place[1] (ID)

[1]*Institute of Sport Sciences, University of Lausanne, Lausanne, Switzerland*
[2]*Nestlé Institute of Health Sciences, Nestlé Research, Lausanne, Switzerland*
[3]*Nestlé Institute of Food Safety & Analytical Sciences, Nestlé Research, Lausanne, Switzerland*

Handling Editors: Paul Greenhaff & Jørn Helge

The peer review history is available in the Supporting Information section of this article (https://doi.org/10.1113/JP290316#support-information-section).

**Abstract figure legend** In a double-blind, crossover study, we observed that olive leaf extract (OLE) intake combined with a single session of moderate-intensity continuous exercise (MICE) significantly enhanced skeletal muscle mitochondrial bioenergetics, whereas this effect was milder when OLE was combined with sprint interval exercise (SIE). Additionally, OLE intake modulated the inflammatory response to both exercise modalities. Figure created with Biorender.com.

**Abstract** Sprint interval exercise (SIE) induces skeletal muscle mitochondrial adaptations that are comparable to, or greater than, those observed with moderate-intensity continuous exercise (MICE), despite requiring a lower training volume. Previous work has shown that these adaptations are at least partly mediated by enhanced mitochondrial bioenergetics, including increased mitochondrial $Ca^{2+}$

uptake and resulting pyruvate dehydrogenase (PDH) activation. In parallel, the natural compound oleuropein from olive leaf extract (OLE) promotes mitochondrial $Ca^{2+}$ uptake and activates PDH in mouse skeletal muscle. Here, we tested the hypothesis that OLE intake would activate PDH and potentiate mitochondrial adaptations in human skeletal muscle during either MICE or SIE. In a crossover, double-blind study, healthy males performed MICE (1 h at 50% maximal aerobic power, $n = 11$) or SIE ($6 \times 30$ s all-out sprints with 4 min recovery, $n = 10$). Knee extensor neuromuscular tests and vastus lateralis muscle biopsies were performed before, immediately after and 24 h after SIE or MICE. OLE improved the decline of power output during the first sprint in SIE and reduced heart rate during MICE but did not affect knee extensor fatigability after both exercise modalities. Transcriptomic analyses revealed an effect of OLE on the mitochondrial and inflammatory response after MICE and SIE, while OLE increased PDH activity in combination with exercise only following MICE. Together, these results suggest that OLE modulates skeletal muscle response to exercise and pave the way for future investigations aiming to investigate the chronic effect of combining OLE and exercise training.

(Received 13 October 2025; accepted after revision 19 March 2026; first published online 15 April 2026)

**Corresponding author** N. Place: Institute of Sport Sciences, University of Lausanne, Quartier UNIL-Centre, Bâtiment Synathlon, 1015 Lausanne, Switzerland. Email: nicolas.place@unil.ch

**Key points**

- Previous studies have shown that oleuropein increases mitochondrial calcium uptake in preclinical models and that mitochondrial calcium uptake contributes to skeletal muscle mitochondrial adaptations in response to maximal intensity exercise in humans.
- Olive leaf extract (OLE) increases the activity of pyruvate dehydrogenase, a proxy of mitochondria calcium uptake, when combined with moderate-intensity exercise.
- Combining moderate-intensity continuous exercise and sprint interval exercise with OLE enhances the mitochondrial response at a transcriptional level.
- OLE enhances skeletal muscle mitochondrial response to acute exercise, paving the way for investigating its effect in combination with chronic exercise training protocols.

## Introduction

Enhancing cardiorespiratory fitness by means of regular aerobic exercise is a clinically proven, cost-effective, primary intervention that can lead to substantial health benefits and can delay or prevent the health burdens associated with many chronic diseases (Myers et al., 2002). The classical paradigm of endurance exercise consists in performing continuous effort at a moderate intensity during a prolonged period of time (moderate-intensity continuous exercise, MICE). While this modality of exercise is effective in improving muscle health and overall performance (Mølmen et al., 2025), a commonly cited limitation for engaging in regular exercise is 'lack of time' (Gibala & Hawley, 2017). This urge for exercise regimens characterized by lower time commitment led to the development of alternative exercise modalities such as sprint interval exercise (SIE). SIE is a form of interval exercise characterized by short bouts of all-out effort interspersed by rest periods that is equally or more effective than traditional MICE for systemic and muscle metabolic adaptations following a short training period, despite a

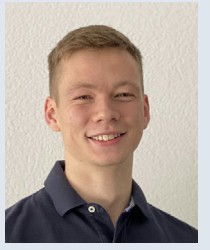

**Clément Lanfranchi** is a PhD student at the Institute of Sport Sciences at the University of Lausanne. His main research interest is skeletal muscle responses to exercise and other stimuli at both functional and molecular levels, using a range of experimental approaches, from human studies to *in vitro* models, with the aim of comprehensively characterizing specific muscle responses. Through this integrative approach, he aims to advance our understanding of how skeletal muscle responds and adapts not only to different exercise stimuli, but also to other factors such as nutritional interventions and environmental constraints.

lower workload and shorter duration (Burgomaster et al., 2008; Gibala et al., 2006; Mølmen et al., 2025).

Activation of mitochondrial energy metabolism contributes to some differences in adaptations to SIE *versus* MICE. It has been reported that the ryanodine receptor 1 (RyR1) was leaky in response to SIE but not MICE in healthy active men, leading to $Ca^{2+}$ release to the cytosol, potentially responsible for some of the beneficial metabolic adaptations previously reported (Burgomaster et al., 2008; Place et al., 2015; Schlittler et al., 2019). Combining exercise studies in humans and in *in vitro* simulated exercise in C2C12 myotubes, we have recently shown that SIE and not MICE increases mitochondrial calcium ($Ca^{2+}$) uptake, leading to pyruvate dehydrogenase (PDH) activation (Zanou et al., 2021). This $Ca^{2+}$-driven pathway led to greater changes in the level of oxidative phosphorylation (OXPHOS) proteins and mitochondrial respiration (Zanou et al., 2021). Despite the powerful effect of intense exercise on improving metabolic function and the increasing prescription of exercise as a therapeutic intervention, any approach aimed at improving the metabolic adaptations induced by either moderate- or high-intensity exercise will be beneficial for both the clinical and general population.

Therefore, we hypothesized that it would be possible to activate PDH during less intense exercise such as MICE and further activate this pathway during SIE. Recently, the natural polyphenol oleuropein from olive leaf extracts (OLEs) was shown to boost muscle mitochondrial energy metabolism and performance in primary human muscle cells and in mice (Gherardi et al., 2025). By directly binding to the MICU1 subunit of the mitochondrial $Ca^{2+}$ uniporter (MCU), oleuropein stimulates mitochondrial $Ca^{2+}$ uptake, activates PDH via its mitochondrial $Ca^{2+}$-dependent dephosphorylation and increases oxidative metabolism (Gherardi et al., 2025). Interestingly, OLE supplementation in mice improved fatigue resistance and endurance performance. As exercise triggers a large release of $Ca^{2+}$ from the sarcoplasmic reticulum to the cytosol, activating the same bioenergetic pathway as oleuropein (Zanou et al., 2021), we reasoned that its combination with this natural bioactive nutrient may increase the concentration of $Ca^{2+}$ microdomains, accelerate mitochondrial $Ca^{2+}$ uptake and improve the mitochondrial bioenergetic response. OLE supplementation in humans increased the levels of circulatory oleuropein metabolites and activated PDH in skeletal muscle of healthy older people, without inducing any functional muscle benefits (Pinckaers et al., 2025). These results suggested that the bioenergetic effects of OLE may manifest under metabolic challenges such as bioenergetic decline during frailty and sarcopenia or increased bioenergetic demands during exercise. Yet it is unknown whether OLE may have beneficial effects in a human exercising setting.

**Table 1. Participant characteristics for each group, MICE and SIE.**

| Characteristic | MICE ($n = 11$) | SIE ($n = 11$) |
|---|---|---|
| Age (years) | $26 \pm 6$ | $26 \pm 5$ |
| Body mass (kg) | $77 \pm 8$ | $78 \pm 6$ |
| Height (cm) | $182 \pm 10$ | $181 \pm 7$ |
| Body mass index (kg/m$^2$) | $23 \pm 2$ | $24 \pm 1$ |
| Weekly physical activity (h) | $6 \pm 3$ | $7 \pm 3$ |
| Maximal aerobic power (W) | $334 \pm 57$ | $370 \pm 51$ |

There was no statistical difference between both groups. MAP: maximal aerobic power.

Given the convergent molecular effects of exercise (Zanou et al., 2021) and OLE on muscle energy metabolism (Gherardi et al., 2025; Pinckaers et al., 2025), the primary aim of this study was to test whether OLE supplementation would cross-talk with exercise and enhance mitochondrial bioenergetic and skeletal muscle performance during an acute session of exercise in healthy untrained young adults. A secondary aim was to investigate whether OLE intake would affect the extent and origin of neuromuscular adjustments induced by the acute exercise session. We hypothesized that combining OLE to SIE or MICE would influence skeletal muscle bioenergetics according to exercise intensity and duration. To test this hypothesis, we conducted a randomized, double-blind, cross-over study on healthy recreationally active young adults who performed a single session of either MICE or SIE in combination with a cross-over of OLE and placebo.

## Materials and methods

### Ethical approval

This study was designed and performed at the University of Lausanne, received approval from the Ethical Commission for Human Research (CER-VD, protocol 2021-00373), adhered to the principles of the *Declaration of Helsinki* and was registered as NCT05350566 on ClinicalTrials.gov. All participants provided written informed consent after being fully informed about the experimental procedures and potential risks.

### Participants

After being informed of the potential risks and discomfort related to the different experimental procedures, 22 young healthy men were randomly allocated to either a MICE or SIE group (Table 1). Participants enrolled had to be physically active without following any structured training programme. Prior to starting the experimentation, all the

participants gave written informed consent and completed a health questionnaire to exclude individuals who might be at risk during the exercise protocol or the biopsy procedure.

### Investigational product description

The active investigational product was an OLE enriched from *Olea europaea* L. and standardized to contain 40% oleuropein (Solabia, Maastricht, The Netherlands) (Pinckaers et al., 2025). Each capsule contained 250 mg of OLE including 100 mg oleuropein, and 139 mg microcrystalline cellulose. Clinical trials support the safety of the dose of oleuropein used in in this paper, with no adverse events reported (Horcajada et al., 2022; Pinckaers et al., 2025). In addition, olive leaf consumption is authorized as a dietary supplement by European countries recognizing the BELFRIT list. The placebo (PLA) capsules were made of 336 mg of cellulose microcrystalline for isovolumetric matching the OLE investigational product capsules. Both the investigational product and the placebo looked and tasted identical and were blinded and coded to participants and investigators.

### Experimental procedures

Before starting data collection (minimum 4 days before the first experimental session), each participant underwent a familiarization session for knee extensor neuromuscular assessments and to measure their maximal aerobic power (MAP). Upon arrival for the first experimental session, participants were given a pill containing either OLE or PLA in a double-blind, randomized manner. Knee extensor neuromuscular assessment and vastus lateralis (VL) muscle biopsies were performed 30 min after ingesting the pill (Pre), immediately after (Post) and 24 h after (24 h) MICE or SIE. The same procedure but with the switched pill

was repeated after a wash-out period of 10–18 days (see Fig. 1). To limit any interferences with the potential effects of the substance, participants were asked to refrain from consuming caffeine/theine on the testing days. They were also asked to eat a similar dinner, breakfast and lunch prior to the sessions as well as to refrain from any strenuous physical activity for 24 h before any experimental sessions. The testing sessions were scheduled at the same time of the day for a given participant (±90 min).

**Familiarization.** The familiarization session was carried out 4–14 days prior to the first testing session to accustom participants to electrical stimulation and voluntary contraction, and to measure their MAP that would later determine the intensity of the MICE. First, the knee extensor neuromuscular assessment was performed following the same protocol as during the experimental sessions (see Neuromuscular assessments). MAP was measured on a cycle ergometer (Lode Excalibur Sport, Lode, Groningen, The Netherlands) through an incremental protocol to exhaustion. The ergometer was first adapted to the participant's morphology. The test started at 0 W and increased by 1 W every 2 s. The participants had to remain seated throughout the entire test. They were recommended to cycle between 60 and 90 rpm and the test was stopped when they could not maintain a cadence of 60 rpm for several seconds. The power reached at the end of the test was considered as the MAP.

**Muscle microbiopsies.** Needle biopsies were taken from the right VL muscle. Briefly, after skin sterilization and local anaesthesia, a 1–2-mm-long skin cut was made with the tip of a scalpel. Biopsies were collected using an automatic biopsy device (Bard Biopsy Instrument, Bard Radiology, Covington, GA, USA). A 14-gauge disposable trocar mounted in the device was inserted through the cut, perpendicular to the muscle fibres, until the fascia

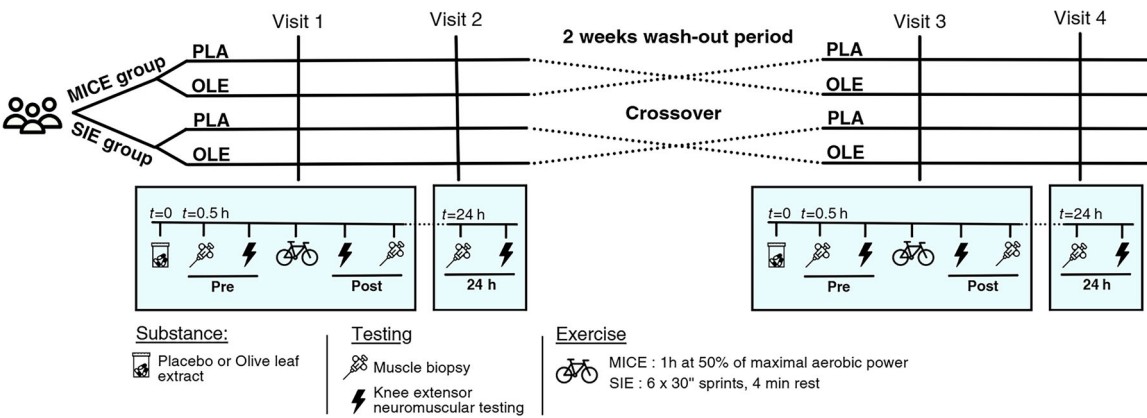

**Figure 1. Graphical representation of the experimental procedures**
MICE, moderate-intensity continuous exercise; SIE, sprint interval exercise.

was pierced. Three or four samples (5–15 mg each) were collected at different angles from one puncture site. Muscle samples were immediately frozen in liquid nitrogen and stored at −80°C until analysis. Biopsies at Pre and Post were collected through the same incision, while for the biopsies at Post 24 h another incision ∼2 cm proximal from the first one was performed. All analyses of the muscle biopsies were conducted between 6 months and 1 year after the completion of data collection.

**Neuromuscular assessments.** A standardized warm-up entailing several submaximal contractions (from 20% to 80% of the maximal estimated strength) was followed by a test battery consisting of a 3 s maximal voluntary isometric contraction (MVC) of the knee extensors with one superimposed doublet at 100 Hz (paired stimuli) [to quantify MVC force, and voluntary activation level (VAL) according to the twitch interpolation technique], followed by one doublet at 100 Hz, one doublet at 10 Hz (to quantify 100 Hz doublet peak force and 10/100 Hz ratio), and a single stimulus [to quantify the muscle compound action potential (M-wave)] delivered at every 2 s after the MVC. Paired and single stimuli of 1 ms were delivered at 400 V through a circular cathode (diameter: 2 cm, same model as electrodes used for EMG recordings) positioned at the femoral triangle level over the femoral nerve with a transcutaneous constant current electrical stimulator (model DS7AH, Digitimer, Welwyn Garden City, UK). The anode (10 × 5 cm, Compex Sa, Ecublens, Switzerland) was placed in the gluteal fold opposite to the cathode. For MVC, participants were asked to raise the force progressively (1–2 s) until a plateau was reached and then maintain this force for 2–3 s. The superimposed paired stimuli were delivered manually during the plateau phase. The investigator provided strong verbal encouragement during the MVC. A supramaximal current intensity (120% of the optimal intensity) was used during the entire experiment to maintain an optimal stimulation of the femoral nerve (Neyroud et al., 2014). The optimal stimulation intensity was assessed incrementally at the beginning of each test session and was reached when a higher intensity of stimulation did not exert a further increase in peak twitch torque and M-wave amplitude.

**Torque and EMG recordings.** Voluntary and evoked knee extension forces were recorded using an isometric ergometer consisting of a custom-built chair equipped with a strain gauge (STS 250 kg, sensitivity 2.0 mV/V and 1.7 mV/N, SWJ, China), sampled at a rate of 1 kHz using an AD conversion system (MP150, BIOPAC, Goleta, CA, USA), and stored for off-line analysis with commercial software (AcqKnowledge, BIOPAC). The knee joint was positioned at an angle of 90° and the trunk–thigh at 100° (180° = full extension). The dynamometer axis

was aligned with the knee extension axis, and the lever arm was fixed to the shank with a strap. The upper body was strapped to the chair with a belt across the abdomen and two cross-shoulder harnesses to limit compensatory movements. Participants were asked to maintain their arms crossed on their chest during the entire assessment. Measurements were made on the same leg (right) throughout the study using the same chair position settings.

The EMG activity of VL was recorded using pairs of circular silver chloride (Ag/AgCl) electrodes (recording diameter: 10 mm, interelectrode distance: 20 mm, Kendall Meditrace 100, Tyco, Cork, Ireland) positioned on the muscle belly according to the SENIAM recommendations (Hermens et al., 2000). The reference electrode was placed over the patella. Low resistance between electrodes was obtained by shaving and cleaning the skin with alcohol. EMG signals were amplified (gain = 1000), filtered through a 10–500 Hz band pass filter, recorded at a sampling rate of 2 kHz, and stored for off-line analysis with the same hardware and software used for the force signal.

**Exercise protocol.** The MICE trial consisted of 1 h of cycling at 50% of the MAP with a 5 min warm-up at 100 W. For this exercise, the participants were recommended to keep a stable cadence between 60 and 90 rpm. The SIE trial consisted of a 4 min warm up at 100 W followed by six repetitions of 30 s all-out cycling bouts at a resistance of 0.7 Nm/kg body mass interspersed by 4 min recovery periods (Zanou et al., 2021). During SIE, the participants were asked to reach their maximum cadence 1–2 s prior to the beginning of the 30 s sprint, and they were verbally encouraged during the entire exercise. In both exercises, participants' heart rate (HR) was monitored (S610i, Polar Electro Oy, Kempele, Finland). Participants were asked to remain seated and were allowed to drink water. They repeated the same exercise trial on the first and the third experimental session.

### Data analysis

**Exercise-related data.** For the MICE group, mean HR was calculated over the whole duration of exercise.

For the SIE trial, we considered HR at the end of each 30 s sprint. In addition, mean power, peak power, work and fatigue index were measured for each sprint performed. The fatigue index was based on the formula given by Naharudin and Yusof (2013) that used the mean power of the first 5 s and last 5 s of each sprint:

$$\text{Fatigue index} = \left[ \frac{(\text{Mean 5 } s \text{ start} - \text{Mean 5 } s \text{ end})}{\text{Mean 5 } s \text{ start}} \right] \times 100.$$

Then, based on previous data, total mean power and total work were averaged over the six repetitions.

**Force and EMG signals.** The MVC force was considered as the highest level of force attained during contraction. The peak force evoked by the 100 Hz doublet (PS100) was quantified to assess contractile properties (Place et al., 2007) and the ratio between the peak force evoked by the 10 Hz doublet and PS100 (PS10/100), used to measure prolonged low-frequency force depression, was calculated as an index of intracellular $Ca^{2+}$ release impairment (Verges et al., 2009; Westerblad et al., 1993).

The twitch interpolation technique was used to evaluate VAL, which was estimated using the following formula (Strojnik & Komi, 1998):

$$VAL = [100 - D \times (MVC_{@stim} \div MVC) \div PS100 \times 100],$$

where $D$ = difference between the force just before the stimulation ($MVC_{@stim}$) and the superimposed doublet amplitude and PS100 = 100 Hz potentiated doublet force at rest. VL M-wave first phase amplitude in response to the single stimulation was measured as an index of sarcolemmal excitability (Rodriguez-Falces & Place, 2018). Due to poor quality of signals, some parameters could not be measured in all participants. When the dataset did not include 11 participants, the specific number of participants used for calculating the parameters is disclosed.

**Western blot analysis.** Muscle samples were suspended in a lysis buffer (100 μL/5 mg of tissue) containing the following: 20 mM Tris/HCl (pH 6.8), 2 mM EDTA (pH 8), 137 mM NaCl, 10% glycerol, 1% Triton X-100, 10 mM beta-glycerophosphate, 1 mM $KH_2PO_4$, 1 mM PMSF, 1 mM $NaVO_3$, 50 mM NaF and a protease inhibitor mixture (Complete Mini, Roche, Basel, Switzerland). The preparation was homogenized with a pestle mounted on its motor, incubated for 1 h at 4°C on a wheel, and sonicated. The lysate was then centrifuged at 9300 *g* for 10 min and the supernatant was transferred into a new tube. Protein concentration was quantified in duplicate for each sample using a BCA kit following the manufacturer's instructions (Thermo Fisher Scientific, Ecublens, Switzerland). Laemmli sample buffer (2×) containing SDS and 2-mercapto-ethanol (Bio-Rad, Hercules, CA, USA) was added to 10 or 20 μg of proteins. Then, samples were electrophoresed for 45 min on 4–15% SDS-precast gradient gels (Bio-Rad) and wet transferred for 1 h onto preactivated PVDF membranes. Total protein quantification was performed using a Revert$^{TM}$ 700 Total Protein Stain Kit (LI-COR, Lincoln, NE, USA) following the manufacturer's instructions and membranes were incubated for 1 h at room temperature with PBS-LI-COR blocking buffer (LI-COR). Then, blots were incubated overnight at 4°C with mouse anti-total OXPHOS (Abcam, Cambridge, UK, 1:1000) diluted in PBS blocking buffer. Membranes were washed in PBS-Tween-20 and incubated for 1 h at room temperature with secondary antibody IRDye 680-conjugated donkey antimouse and IRDye 800-conjugated donkey anti-rabbit IgG (LI-COR, 1:10,000 and 1:5000, respectively) or IRDye 680-conjugated goat anti-mouse and IRDye 800-conjugated goat anti-rabbit IgG (LI-COR, 1: 100,000 and 1:10,000, respectively). Immunoreactive bands were visualized using infrared fluorescence (IR-Odyssey scanner, LI-COR), and band densities were quantified using Image Studio v.5.2.5 (LI-COR). Protein intensity signal was normalized to that of total protein staining intensity signal. Each target protein quantification was performed in duplicate, and the average was expressed as a percentage of the Pre-PLA values.

**PDH activity assay.** Muscle PDH activity was measured as previously described (Constantin-Teodosiu et al., 1991) with slight modifications (i.e. use of a commercially available PDH assay kit, as explained below). Briefly, ∼10 mg wet muscle tissue was used for the determination of PDH activity. All muscle samples were homogenized in 200 μL of ice-cold homogenization buffer using a motor-drive pestle although kept on ice. The homogenization buffer contained 200 mM sucrose, 50 mM KCl, 5 mM $MgCl_2$, 5 mM EGTA, 50 mM Tris, 50 mM NaF, 5 mM dichloroacetate (DCA) and 0.1% (v/v) Triton X-100 (pH 7.8). NaF was added in the homogenization buffer as a general phosphatase inhibitor to prevent PDH dephosphorylation. EGTA and DCA are commonly included in PDH activity measurement protocols during tissue or cell homogenization to inhibit PDH dephosphorylation by the $Ca^{2+}$-sensitive pyruvate dehydrogenase phosphatases (PDP) and phosphorylation by pyruvate dehydrogenase kinases (PDK), respectively (Kerr et al., 2012). Using this standard assay thus ensures that PDH activity is measured in a situation where its *in vivo* phosphorylation state is preserved. After homogenization, the final volume was adjusted to 30 μL of homogenization buffer/milligram of wet muscle. PDH activity was measured spectrophotometrically (450 nm, 37 °C) using a commercially available kit and according to the manufacturer's instruction (MAK183; Sigma-Aldrich). Before analysis, the samples were centrifuged at 400 *g* for 5 min to spin down any insoluble material. The assay was performed according to the manufacturer's instructions using 5 μL of muscle homogenate, with optical density measured every 3 min during 90 min. The activity of PDH was calculated as the change in NADH concentration over time per milligram of wet muscle using a standard curve.

## Transcriptomics

*RNA extraction.* Approximately 10 mg wet muscle was lysed with 400 μL of lysis buffer (Agencourt RNAdvance Tissue Kit, Beckman Coulter, Indianapolis, IN, USA) in a Lysing Matrix D tube (MP Biomedicals, Santa Ana, California, USA). Subsequent homogenization of the samples was performed using a bead beater (FastPrep-24™, MP Biomedicals) for two cycles of 60 s at speed 6. Total RNA was then extracted using the Agencourt RNAdvance Tissue Kit (Beckman Coulter) according to the provider's recommendations. Total RNA was quantified using the Quant-iT™ RiboGreen™ RNA Assay Kit (Invitrogen, Carlsbad, CA, USA) on a Spectramax M2 (Molecular Devices, Sunnyvale, CA, USA). Total RNA quality assessment was done using Fragment Analyzer-96 with DNF-471-0500 Standard Sensitivity RNA Analysis Kit (Agilent Technologies, Santa Clara, California, USA).

*mRNAseq library preparation and sequencing.* An total of 300 ng of total RNA was used of each RNA sample. A bead-based mRNA capture followed by a fragmentation, first- and second-strand synthesis, an end repair and a 3′ adenylation and indexed linker ligation was done. The adapter ligated cDNA library fragments were enriched by PCR using 13 cycles of amplification, bead purified and quantified by Qubit HS DNA kit (Invitrogen). Size pattern was controlled with Fragment Analyzer-96 with DNF-474-0500 High Sensitivity NGS Fragment Analysis Kit (Agilent Technologies). Libraries were pooled at an equimolar ratio and clustered at a concentration of 650 pmol on a paired end sequencing flow cell. Two times 59 cycles of sequencing were performed on an Illumina NextSeq2000 on six P3 100 kits (Illumina Inc., San Diego, CA, USA). Sequencing raw data (bcl) were demultiplexed on the instrument with Dragen v.3.10.11. Reads were aligned to the human genome (GRCh38-101) using STAR v.2.7.3a (5). The number of uniquely mapped reads was between $3.3 \times 10^7$ and $5.5 \times 10^7$ with an average of $4.7 \times 10^7$.

*mRNA differential expression and pathway enrichment analyses.* All statistical analyses were performed using R v.4.2.2 and relevant *Bioconductor* packages (e.g. *limma* 3.54.2, *edgeR* 3.40.2). Unless otherwise stated, 10 participants undergoing SIE exercise and 10 participants undergoing MICE exercise were analysed, for a total of 119 muscle biopsies (for 19 participants we could sequence six muscle biopsies, and for one participant we could only sequence five muscle biopsies). The acute effect of the intervention before exercise was estimated using all 20 participants. For differential expression analysis, all available muscle biopsies were included. Differentially expressed genes due to the exercise effect and to the intervention effect at different time points were defined using the *limma* package (Smyth, 2004).

Briefly, after removing low-expressed genes using the *edgeR* function 'filterByExpr' with default parameters and specifying the muscle biopsy groups, data were normalized by the trimmed mean of M-values method as implemented in the *edgeR* function 'calcNormFactors' (Robinson et al., 2010), and the 'voomWithQualityWeights' function was applied to model the mean-variance relationship and estimate the sample-specific quality weights (Liu et al., 2015). To account for the longitudinal nature of the dataset, correlations between repeated measurements from the same donor were estimated using the 'duplicateCorrelation' method (Smyth et al., 2005), which was estimated twice. *P*-values were corrected for multiple testing using the Benjamini–Hochberg method.

Pathway enrichment analysis was performed using CAMERA (Wu & Smyth, 2012), a competitive gene set test querying whether a set of genes annotated in the Molecular Signatures Database (MSigDB) (Subramanian et al., 2005) is enriched. Hallmark gene sets from MsigDB and gene ontology biological processes from MSigDB, C5 (GO BP gene sets), were accessed using the R library msigdbr 7.5.1 and used to perform pathway analyses. Given the great interest in genes coding for proteins with mitochondrial localization, sub-mitochondrial compartmentalization and the associated pathways, we also used Human MitoCarta 3.0 datasets (Rath et al., 2021).

The *Mfuzz* package (Kumar & E. Futschik, 2007) was used for soft clustering of all significantly regulated genes in MICE and SIE groups and included the use of a *c*-value of 4, and an *m*-value of 3.68 that was derived from the 'mestimate' function. Soft cluster images were created with the 'mfuzz.plot2' function, and the 'acore' function was used to extract the cluster membership scores for each gene.

## Proteomics

*Sample preparation.* Protein extraction was performed using an extraction buffer containing RIPA buffer, PhoSTOP and protease inhibitors cocktail (Roche). Muscle samples ∼10 mg in 150 μL of extraction buffer were homogenized using a Fast-Prep-24 homogenizer (MP Biomedicals) at a speed of 6.0 m/s for 40 s, with a cooling period of 1 min on ice between two cycles. The samples were then centrifuged at 14,000 *g* for 20 min at 4°C to remove cellular debris. Protein concentration was determined using the Bradford assay (Bio-Rad) following the manufacturer's instructions.

Sample volumes corresponding to 100 μg proteins were pipetted into a 96-well plate and dried in a SpeedVac concentrator (Thermo Scientific). With the exception of

the acetone precipitation, the sample preparation was performed on a four-channel Microlab Star liquid handler (Hamilton, Bonaduz, Switzerland) as previously described (Pedersen et al., 2025). Briefly, the samples were diluted in 100 μL of 100 mM triethylammonium hydrogen carbonate buffer (TEAB) in $H_2O$ supplemented with 0.007 μg/μL bovine $\beta$-lactoglobulin (BLG), reduced with 5.3 μL of 200 mM tris (2-carboxyethyl) phosphine hydrochloride in $H_2O$ and alkylated with 5.5 μL of 375 mM iodoacetamide in 100 mM TEAB. Proteins were precipitated with 600 μL cold acetone for 1 h at –20°C. The samples were centrifuged for 20 min at 4000 rpm at 4°C. The supernatant was removed before the remaining pellet was dried in a SpeedVac concentrator. Protein pellets were resuspended in 100 μL of 100 mM TEAB, digested overnight at 37°C with 10 μL of 0.25 μg/μL trypsin/Lys-C and 6-plex tandem mass tag (Thermo Scientific) labelled with 41 μL reagent at 18.5 μg/μL in $CH_3CN$. The reaction was quenched with 8 μL of 5% hydroxylamine in $H_2O$ and samples were pooled by sets of six differentially labelled samples. Samples were desalted with reversed-phase (RP) solid-phase extraction (Oasis HLB) and purified with SCX (Strata-X-C 33u).

*Liquid chromatography tandem mass spectrometry (LC-MS/MS).* Dried samples were dissolved in 1.5 mL $H_2O/CH_3CN$/formic acid (FA) at 97:3:0.1 and a volume of 3 μL of sample was injected per analysis. Samples were analysed in quadruplicate with RP-LC-MS/MS using a Vanquish NEO UHPLC system coupled to an Orbitrap Fusion Lumos Tribrid mass spectrometer (Thermo Scientific). Briefly, peptides were trapped on an Acclaim PepMap 300 μm × 5 mm (C18, 5 μm, 100 Å) precolumn then eluted on an Acclaim PepMap RSLC 75 μm × 50 cm (C18, 2 μm, 100 Å) nano column (Thermo Scientific) heated to 50°C using a PRSO-V1 column oven (Sonation, Biberach, Germany). Peptide separation was obtained at a flow rate of 300 nL/min over 180 min with the following gradient of mobile phase A ($H_2O/CH_3CN$/FA 98:2:0.1) and B ($H_2O/CH_3CN$/FA 20:80:0.1): from 6.3% (hold for 2 min) to 17.5% B over 85 min, from 17.5% to 25.5% B over 42 min and from 25.5% to 40% B over 28 min followed by washing (98% B) and equilibration (6.3% B) for a further 23 min.

MS data were acquired using data-dependent acquisition in positive mode with an ion spray voltage of 1900 V and transfer tube temperature of 275°C. For the MS survey scans in profile mode, the Orbitrap resolution was set to 120,000 at $m/z = 200$ [automatic gain control (AGC) target of $2 \times 10^5$) with a $m/z$ scan range from 300 to 1500, RF lens set at 30% and maximum injection time at 100 ms. MS/MS was performed with higher-energy collisional dissociation (HCD) at 35% of the normalized collision energy, AGC target was set to $1 \times 10^5$ (isolation width of 0.7 in the quadrupole), with a resolution of 30,000 at $m/z = 100$, and a maximum injection time of 105 ms with Orbitrap acquiring in profile mode. A duty cycle of 3 s was used to determine the number of precursor ions to be selected for HCD-based MS/MS and dynamic inclusion was set for 60 s within a±10 ppm window. A lock mass of $m/z = 445.12002$ was used during all acquisitions.

*Data treatment.* Identification of proteins was performed against the human UniProtKB/Swiss-Prot database (*Homo sapiens* 06/2023 release) including the BLG sequence (20,424 entries in total). The search engine Mascot (v.2.8.2, Matrix Sciences, London, UK) was used via the Mascot Daemon 2.8.0 (Perkins et al., 1999) interface with ProteoWizard msConvert 3.0 for data import (Chambers et al., 2012). Carboamidomethylation of cysteine and 6-plex TMT-labelled lysine (229.163 Da) were set as fixed modifications while oxidized methionine, deamidated asparagine/glutamine and 6-plex TMT-labelled peptide amino terminus (+229.163 Da) were set as variable modifications. The enzyme was set to trypsin, with a maximum of two potential missed cleavages. Peptide ion tolerance was set to 10 ppm and fragment ion tolerance to 0.02 Da.

The resulting Mascot files were loaded into Scaffold Q+ 5.0.0 (Proteome Software, Portland, OR, USA) for further search and validation with X! Tandem [The GPM, thegpm.org; v.Alanine (2017.2.1.4)]. Peptide and protein false discovery rate (FDR) were fixed at maximum 1% with a two-unique-peptide criterion to report protein identification. Isotopic purity corrections of the TMT labels were applied according to the manufacturer's information and quantification was performed with no filter and normalization set to on. The results were exported as $\log_2$ of the protein ratio fold change.

*Protein differential expression and pathway enrichment analyses.* Differentially quantified proteins due to the exercise effect and to the intervention effect at different time points were defined using the *limma* package (Smyth, 2004), after quantile normalization. To account for the longitudinal nature of the dataset, correlations between repeated measurements from the same donor were estimated using the 'duplicateCorrelation' method (Smyth et al., 2005) that was estimated twice. *P*-values were corrected for multiplicity testing using the Benjamini–Hochberg method. Pathway enrichment analysis was performed using Perseus software (Tyanova & Cox, 2018). Proteins were annotated using gene ontology biological processes from MSigDB, C5 (GO BP gene sets), (Subramanian et al., 2005) and the Human MitoCarta 3.0 datasets (Rath et al., 2021). Pathways with fewer than five associated proteins were filtered out. The difference of means obtained from the differential analysis was used for 1D enrichment analysis (Cox & Mann, 2012).

## Statistical analysis

Normality of the data was checked using the Shapiro–Wilk test. To compare the sprint effect (1st *vs.* 2nd *vs.* ... *vs.* 6th sprint) in the two different conditions (PLA *vs.* OLE), linear mixed models with sprint and condition as fixed effects and participant as a random effect were used. When a significant effect was found, appropriate multiple comparisons were conducted to test for differences among pairs of means using *post hoc* tests, corrected with the Bonferroni method. For all paired comparisons (PLA *vs.* OLE), paired *t*-tests were used. To compare the time effect (Pre *vs.* Post *vs.* 24 h) in the two different conditions (PLA *vs.* OLE), linear mixed models with time and condition as fixed effects and participant as a random effect were used. When a significant effect was found, appropriate multiple comparisons were conducted to test for differences among pairs of means using *post hoc* tests, corrected with the Bonferroni correction method. The variation in the *n* values from the initial inclusion is due to the following issues. For performance data of the SIE group, the recording of one participant was not correctly saved. After the first microbiopsy, one participant from the MICE group decided to stop the biopsy procedures but he was still included for the other parameters. One participant from the MICE group was removed from the western blot analyses due to poor quality of the sample and incoherent protein levels. For the PDH activity, measures of two samples were out of the standard curve and were therefore excluded. For all analyses except transcriptomics and proteomics, Jamovi software (v.2.3, Sydney, Australia) was used. The significance level was set at $P \leq 0.05$ or otherwise stated. In the text, data are presented as means ± SD. In graphs, data are presented as means with individual values.

## Results

### Performance and HR during MICE and SIE

During prolonged moderate low-intensity exercise such as MICE, HR adapts to skeletal muscle demands in nutrients and oxygen according to contraction and bioenergetic efficiency. The mean HR during MICE tended to be lower by 3 ± 5 bpm when participants ingested OLE compared to PLA, although this difference did not reach statistical significance ($P = 0.0547$, Fig. 2*A*). During SIE, the mean power gradually decreased from the second sprint without differences between PLA and OLE with a total decrease of $-28.5 \pm 9.4\%$ and $-34.5 \pm 10.2\%$, respectively (main sprint effect: $P < 0.001$ Fig. 2*B*, *C*). The fatigue index increased gradually from the second sprint until the last sprint (main sprint effect: $P < 0.001$, Fig. 2*D*). When considering the first sprint only as an index of anaerobic performance, as a large proportion of ATP resynthesis is supplied by anaerobic metabolic pathways (Smith & Hill, 1991), we

observed that the fatigue index was 25.5 ± 29.1% lower with OLE ingestion as compared to PLA ($P = 0.013$, Fig. 2*E*) without any differences regarding the initial power output (11.0 ± 1.05 *vs.* 10.4 ± 0.98 W/kg for PLA and OLE respectively, $P = 0.15$), suggesting that OLE acutely primes skeletal muscle performance. Finally, the HR measured at the end of each sprint was lower when ingesting OLE as compared to PLA (main group effect: $P = 0.002$). Together, these results suggest that OLE reduces HR during exercise and improves fatigue resistance during the early stage of SIE.

### Neuromuscular function before and after MICE and SIE

To explore whether the effect of OLE modulates the origin and extent of muscle fatigability induced by MICE and SIE, we assessed knee extensor function before and after the exercise. The extent of muscle fatigability investigated through the MVC force loss was similar immediately after MICE and SIE (main time effect: $P < 0.001$) but recovered 24 h after the end of exercise (main time effect: $P < 0.001$, Fig. 3*A*, *B*). Intriguingly, in the MICE group, MVC values were lower with OLE as compared to PLA (main substance effect: $P = 0.043$, Fig. 3*A*) but this was not the case in the SIE group (Fig. 3*B*). The overall fatigue level observed immediately following MICE and SIE may not be explained by central (ne) alterations as VAL remained stable over time (Fig. 3*C*, *D*). We observed no changes in VL M-wave first phase amplitude following MICE and SIE (Fig. 3*E*, *F*), indicating that sarcolemmal excitability was not affected by exercise. However, processes occurring below the action potential propagation along the sarcolemma were affected by both types of exercise (Fig. 3*G*–*J*). First, we observed a decrease in 100 Hz doublet force, an index of peripheral fatigue, immediately after MICE and SIE (main time effect: $P < 0.001$) which recovered 24 h later (main time effect: $P < 0.001$, Fig. 3*G*, *H*). In addition, we observed prolonged low-frequency force depression immediately following MICE and SIE (main time effect: $P < 0.001$) which recovered 24 h later (main time effect: $P < 0.001$, Fig. 3*I*, *J*), indicating alteration in intracellular $Ca^{2+}$ handling during contraction (Chin et al., 1997; Verges et al., 2009; Westerblad et al., 1993). However, OLE did not modulate any of these alterations. Overall, while both exercise modalities impaired muscle force generating capacity, OLE did not modify this response.

### Transcriptional and proteomic response to a session of MICE or SIE

To elucidate the acute molecular response of skeletal muscle following exercise, we performed transcriptomics

and proteomics at investigated time points. MICE significantly changed the mRNA level of 3883 genes at Post and 3608 genes at 24 h (Fig. 4*A*, FDR < 0.05). SIE changed the mRNA level of 190 genes at Post and 2967 genes at 24 h (Fig. 4*B*, FDR < 0.05). Venn diagrams show that only a minority of differentially expressed genes in response to MICE (Fig. 4*C*) and SIE (Fig. 4*D*) are common between the acute (Post) and delayed (24 h) response. Among all the differentially expressed genes, cluster analysis on the response kinetics highlighted different patterns in the transcriptional response (Fig. 4*E*, *G*), indicating that the transcriptomic response to exercise is time dependent, with many genes

being still regulated at later time points such as 24 h following exercise.

Proteomic analysis allowed identification of 1497 proteins. MICE modified the level of 174 proteins at Post and four proteins at 24 h (Fig. 5*A*, FDR < 0.1). SIE modified the level of 15 proteins at Post without any detected changes at 24 h (Fig. 5*B*, FDR < 0.1). To investigate the pathways affected by exercise, we performed an enrichment analysis using the GOBP database (Fig. 5*C*, *D*). The most affected pathways immediately after MICE and SIE were associated with immune and inflammatory responses, while mitochondria-related pathways ('mitochondrial trans-

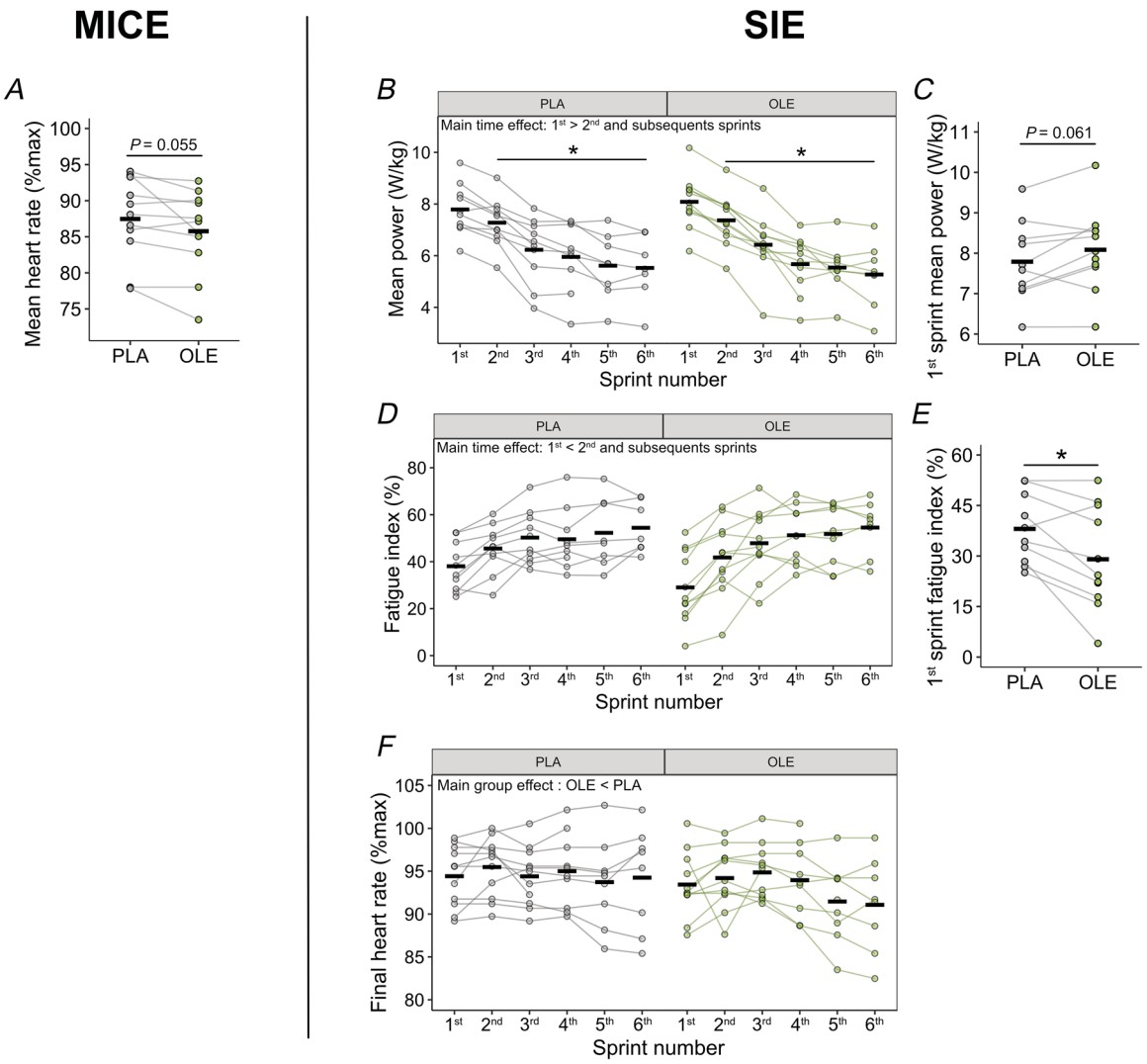

**Figure 2. Heart rate and performance during exercise**

*A*, mean heart rate measured during MICE with ingestion of placebo (PLA) or olive leaf extract (OLE) (*n* = 11). *B*, mean power produced during each sprint; and *C*, mean power produced during the first sprint when combined with PLA or OLE (*n* = 10). *D*, fatigue index for each sprint; and *E*, fatigue index for the first sprint with ingestion of PLA or OLE (*n* = 10). *F*, final heart rate measured at the end of each sprint with ingestion of PLA or OLE (*n* = 11). Black thick lines represent the mean and dots linked with lines represent individual participants. *$P \leq 0.05$.

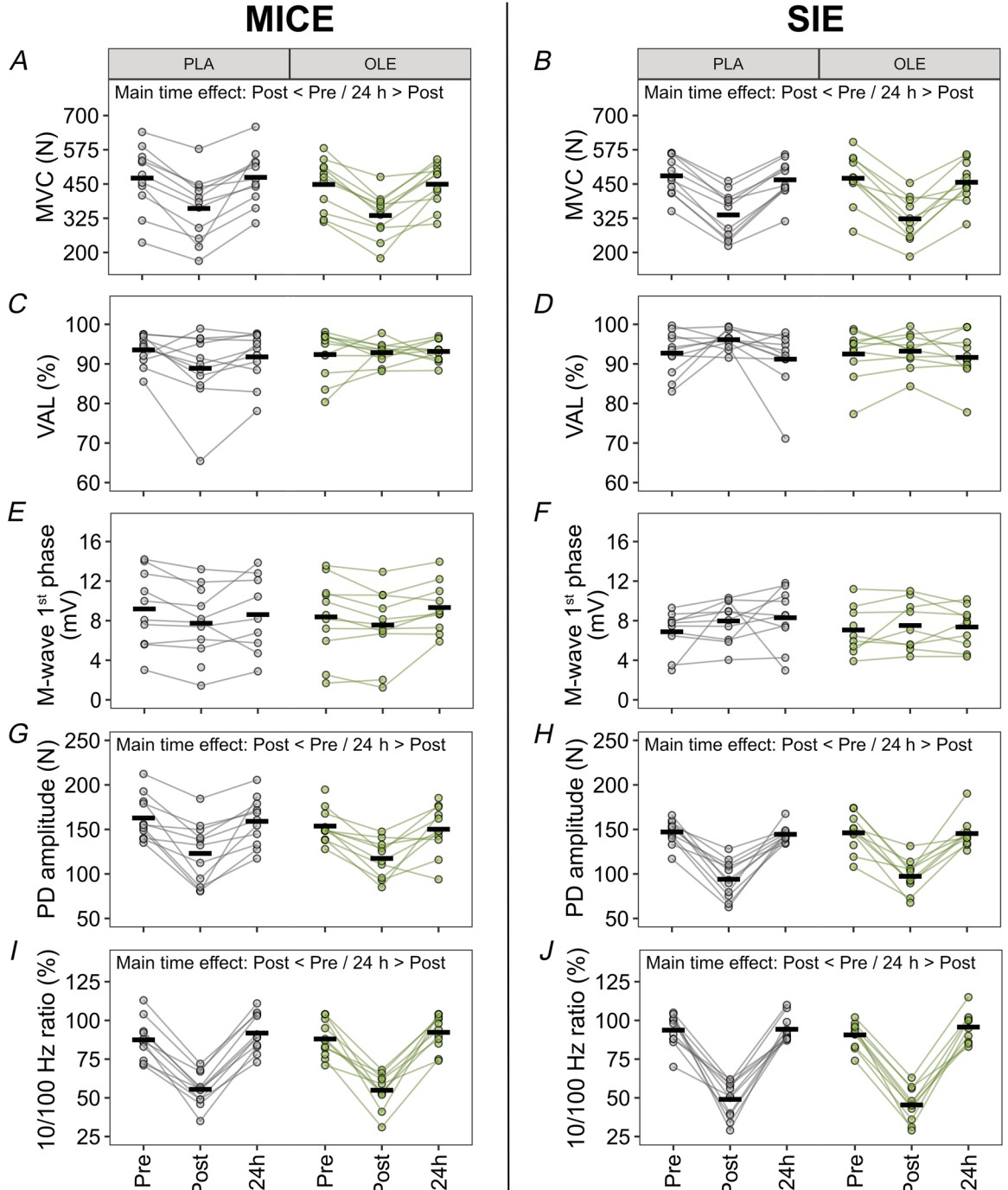

**Figure 3. Knee extensor neuromuscular function assessment before and after MICE and SIE**
*A* and *B*, maximal voluntary contraction force (MVC); *C* and *D*, voluntary activation level (VAL); *E* and *F*, VL M-wave first phase amplitude; *G* and *H*, potentiated doublet (PD) amplitude; and *I* and *J*, 10/100 Hz ratio. Measurements were performed before (Pre), immediately after (Post) and 24 h after (24 h) a session of moderate-intensity continuous exercise (MICE, *n* = 11) or sprint interval exercise (SIE, *n* = 11) with ingestion of placebo (PLA) or olive leaf extract (OLE). Black thick lines represent the mean and dots linked with lines represent individual participants. *$P \leq 0.05$.

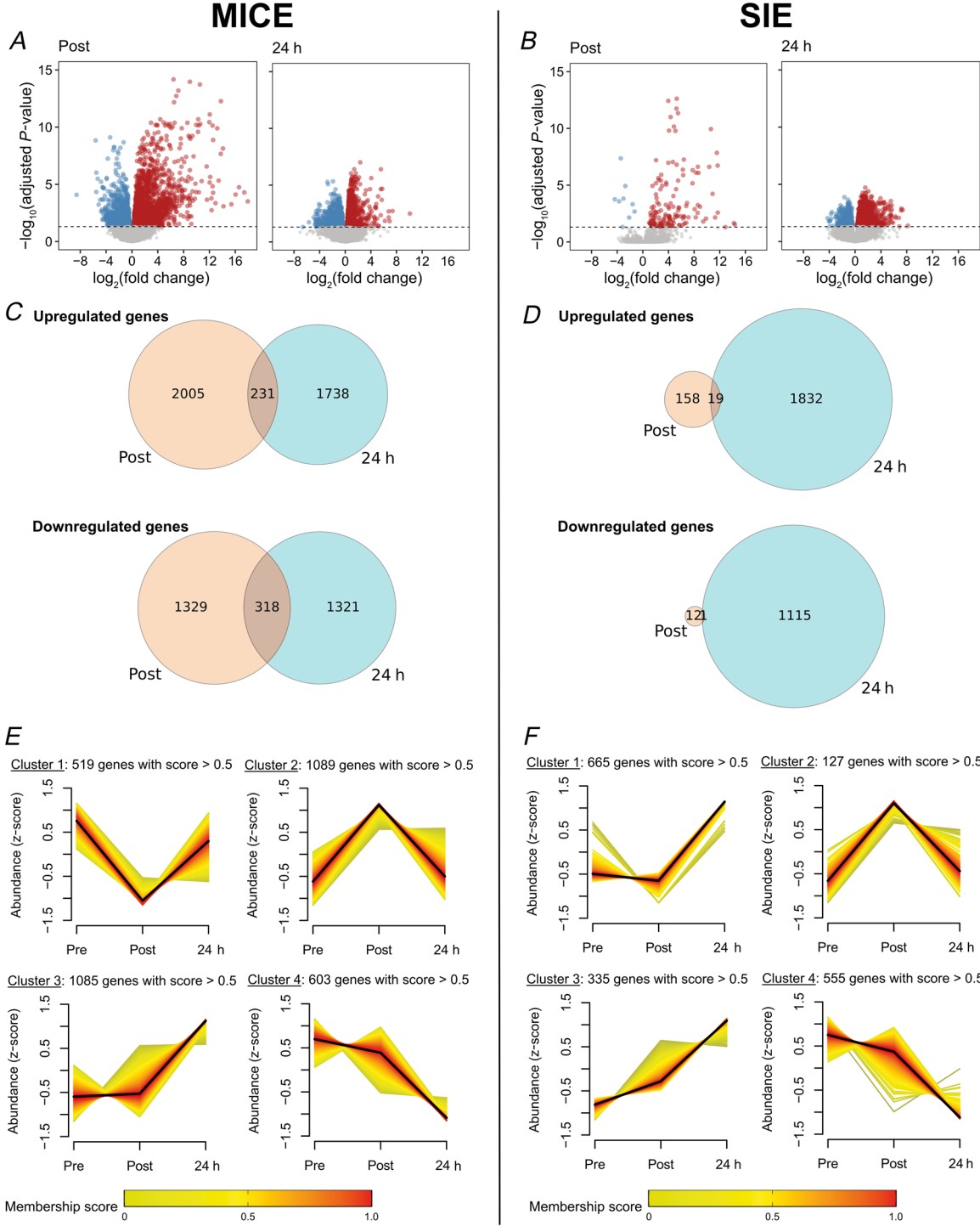

**Figure 4. Transcriptional response of skeletal muscle following MICE and SIE**

*A* and *B*, volcano plot showing an overview of the response to moderate-intensity continuous exercise (MICE, *A*) and sprint interval exercise (SIE, *B*) of all identified genes with placebo ingestion. Blue and red dots indicate significantly down- and up-regulated genes respectively (FDR ≤ 0.05). *C* and *D*, Venn diagrams showing the number of common and distinct significantly modified genes between Post and 24 h after MICE (*C*) and SIE (*D*). *E* and *F*, soft clustering of significantly modified genes reveals four main clusters of response kinetic following MICE (*E*) and SIE (*F*). For all panels, data were obtained from 10 participants.

port' and 'cellular respiration') were rather downregulated at Post, in line with the reported decrease in mitochondrial respiratory capacity following high-intensity exercise (Layec et al., 2018).

### Effects of OLE on the molecular responses to MICE and SIE

As oleuropein has been reported to be an antioxidant with benefits that extend beyond mitochondria (Erten et al., 2016; Mirsanei et al., 2023; Qabaha et al., 2018), we first performed a general enrichment pathway analysis using the HALLMARK database on the RNAseq data. Immediately after MICE, OLE potentiated the acute inflammatory response of TNFα, IL-6 and interferon pathways (Fig. 6*A*) that are described to mediate some of the beneficial effects of exercise adaptation (Merry & Ristow, 2016; Wyckelsma et al., 2026). This effect was transient and resolved within 24 h as the response of these pathways 24 h after MICE was similar or even decreased in participants consuming OLE. Immediately following SIE, OLE had no detectable effect on the acute inflammatory response; however, at 24 h following exercise, OLE attenuated this response and suppressed proteostasis-associated pathways, including mTOR signalling and the unfolded protein response (Fig. 6*B*).

### Mitochondrial responses of OLE ingestion combined with MICE and SIE

To better understand the mitochondrial response to exercise and OLE, we next investigated the specific mitochondrial-related gene expression after exercise and in combination with OLE. First, to check the general transcriptional response related to mitochondria, we used the enrichment analysis using the HALLMARK annotation and specifically checked the 'OXIDATIVE PHOSPHORYLATION' pathway (Fig. 7*A*, *B*). MICE alone up-regulated this pathway enrichment at 24 h (FDR = 0.047) and OLE potentiated this response (FDR < 0.001, Fig. 7*A*). SIE downregulated this pathway at Post (FDR < 0.001) and at 24 h (FDR = 0.022) but OLE strongly increased OXPHOS gene expression at both time points (FDR < 0.001, Fig. 7*B*). To better capture alterations of specific mitochondrial pathways, we then further analysed these mitochondrial signatures using the MitoCarta database (Fig. 7*C*, *D*). After MICE, OLE further up-regulated specific pathways such as 'Mitochondrial ribosome' and 'Translation' that were already upregulated in response to exercise alone. However, it also increased pathways that were not affected by exercise alone such as OXPHOS subunit-related genes (Fig. 7*C*). Most of the mitochondrial pathways were reduced by SIE alone, and strongly increased by OLE, especially after 24 h of

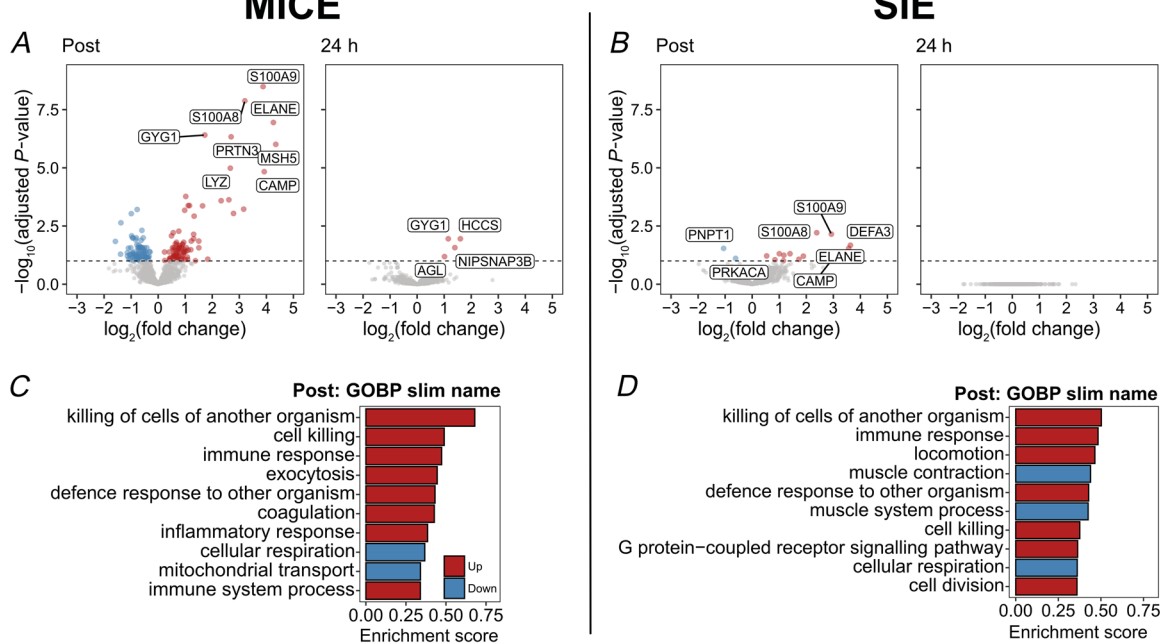

**Figure 5. Effect of MICE and SIE on skeletal muscle proteome**
*A* and *B*, volcano plots showing an overview of the response to moderate-intensity continuous exercise (MICE, *A*) and sprint interval exercise (SIE, *B*) of all identified proteins. Blue and red dots indicate significantly down- and up-regulated proteins, respectively (FDR ≤ 0.05). *C* and *D*, bar plots representing the enrichment of proteomics pathways from the GOBP database following MICE (*C*) and SIE (*D*) combined with placebo. For all panels, data were obtained from 10 participants.

response (Fig. 7D). Overall, these results indicate that OLE consumption 24 h after MICE and SIE enhanced the mitochondrial pathways that were already activated by exercise and additionally stimulated some pathways that had not been activated by exercise alone.

As SIE was reported to increase levels of OXPHOS proteins 24 h after exercise (Zanou et al., 2021), we investigated the protein levels of selected subunits of the electron transport system complexes by western blot. Immediately after MICE, protein levels of CIII, IV and ATPase subunits were decreased (main time effect: $P < 0.001$) and returned to baseline 24 h later (main time effect: $P < 0.001$) without any effect of OLE ingestion compared to PLA (Fig. 8A). Regarding SIE, no alterations were observed for any complexes and OLE ingestion had no effects (Fig. 8B). As the investigation of one subunit per complex may not comprehensively represent the mitochondrial-related response, we also annotated the proteomic data with the MitoCarta database and performed an enrichment analysis. In MICE, it confirmed the acute alteration of mitochondrial proteins observed by targeted western blotting but revealed overall broader mitochondrial alterations (Fig. 8C). SIE also induced a general down-regulation of mitochondrial

proteins at Post, which partially persisted up to 24 h and was potentiated by OLE at both Post and 24 h (Fig. 8D). Overall, the proteomic and targeted analyses revealed a down-regulation of mitochondrial-related proteins immediately after MICE and SIE. Notably, OLE further amplified the decrease in mitochondrial pathways following SIE but not MICE.

### PDH activity following MICE and SIE combined with OLE

Since OLE activates bioenergetics by stimulating PDH activity via increased mitochondrial $Ca^{2+}$ uptake during contraction (Gherardi et al., 2025), we measured PDH activity as a readout of the effect of OLE on mitochondrial $Ca^{2+}$ uptake. A different response was observed between PLA and OLE immediately following MICE, with OLE inducing a faster activation of PDH (interaction effect: $P = 0.0087$, Fig. 9A). PDH activity was increased following SIE with both PLA and OLE ingestion (main time effect: $P < 0.001$, Fig. 9B). One participant showed a substantial increase in the Post measurement following placebo intake (Fig. 9B). To determine whether the main effect of time was driven by this individual, we performed the statistical

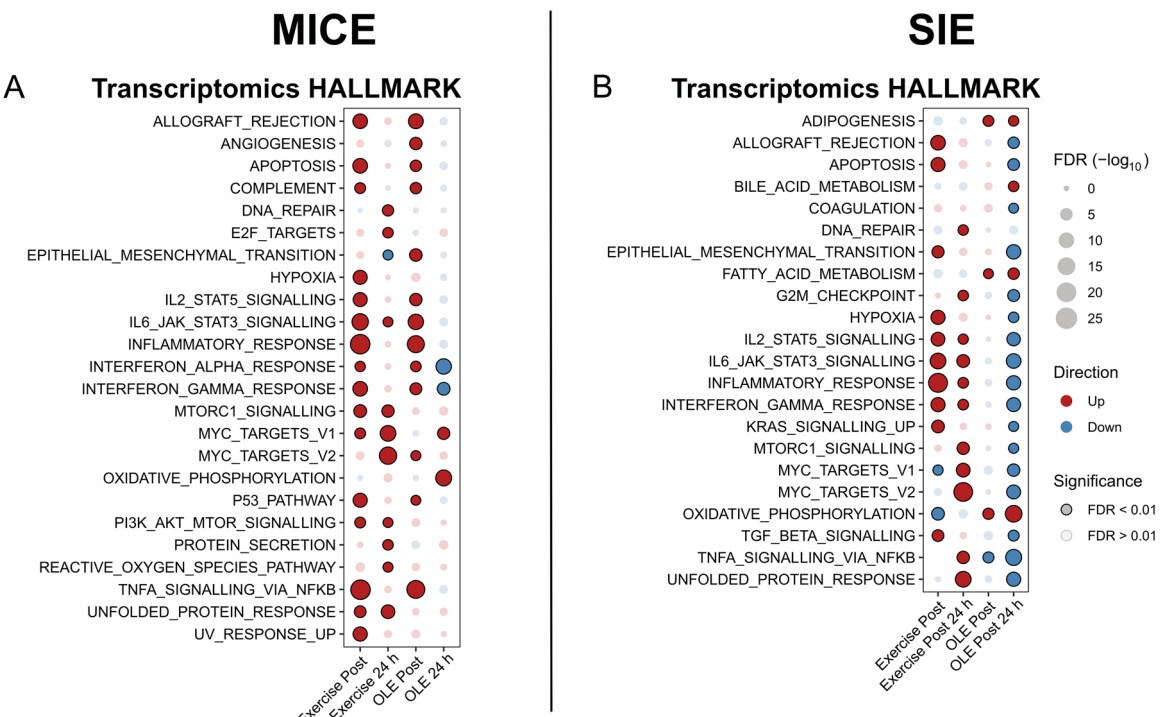

**Figure 6. General effect of olive leaf extract (OLE) combined with exercise on skeletal muscle transcriptome**

*A* and *B*, dot plots representing the enrichment of transcriptomic pathways from the HALLMARK database following moderate-intensity continuous exercise (MICE, *A*) and sprint interval exercise (SIE, *B*) with ingestion of placebo (PLA) or OLE. Columns represent the effect of exercise + PLA at Post (Exercise Post) and at 24 h (Exercise 24 h) and effect of OLE corrected for the effect of exercise at Post (OLE post) and at 24 h (OLE 24 h). For all panels, data were obtained from 10 participants.

analysis after excluding their data. This resulted in a *post hoc* comparison between Pre and Post measurements with a *P* value of 0.055. To better characterize differences between both interventions, we also compared the Pre to Post exercise changes within participants and observed a stronger increase in PDH activity with OLE as compared to PLA after MICE (*P* = 0.0059, Fig. 9*C*), while OLE did not modify the acute increase of PDH activity after SIE (Fig. 9*D*). Overall, these findings indicate an additional effect of OLE intake on PDH activation in response to MICE but not to SIE.

## Discussion

A recent preclinical study revealed the effect of oleuropein, a natural bioactive polyphenol from olive leaves, on mitochondrial $Ca^{2+}$ uptake (Gherardi et al., 2025). Mitochondrial $Ca^{2+}$ uptake has been shown to be critical

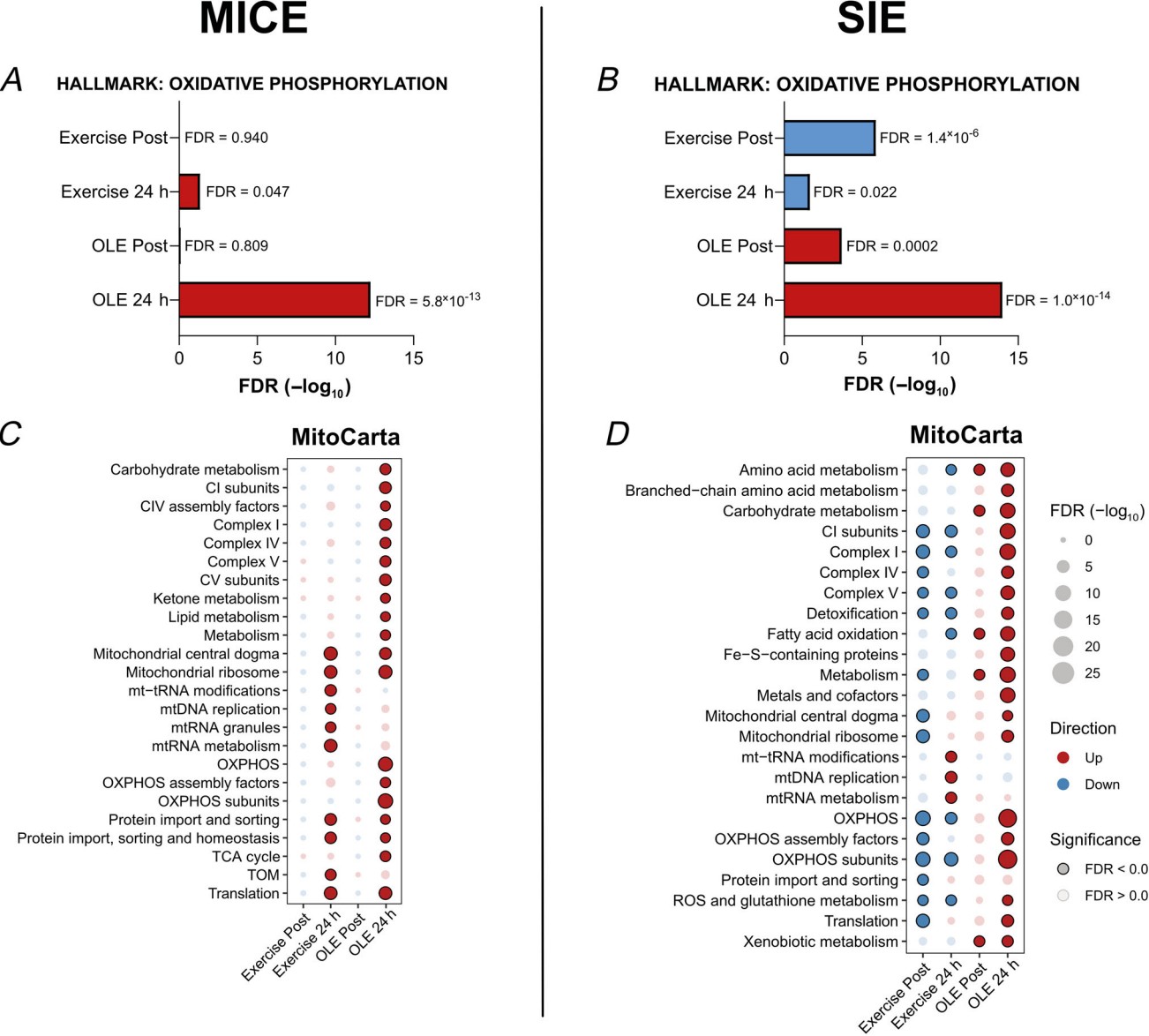

**Figure 7. Olive leaf extract (OLE) effect on mitochondrial-related transcriptome following exercise**
*A* and *B*, enrichment analysis of the HALLMARK 'OXIDATIVE PHOSPHORYLATION' transcriptomic pathway following moderate-intensity continuous exercise (MICE, *A*) and sprint interval exercise (SIE, *B*) when combined with placebo (PLA) or OLE ingestion. Lines represent the effect of exercise + PLA at Post (Exercise Post) and at 24 h (Exercise 24 h) and effect of OLE isolated from exercise at Post (OLE post) and at 24 h (OLE 24 h). *C* and *D*, dot plots representing the enrichment of transcriptomic pathways from the MitoCarta database following MICE (*C*) and SIE (*D*) with ingestion of PLA or OLE. Columns represent the same effect as lines in *A* and *B*. For all panels, data were obtained from 10 participants.

in inducing mitochondrial adaptations following SIE, while being more mildly triggered by MICE (Zanou et al., 2021). Here, we tested two main hypotheses: combining OLE with exercise would improve the mitochondrial response by (I) triggering PDH activation during MICE and (II) by further stimulating PDH activation during SIE, which may improve exercise performance. Our main findings indicate that PDH activity, a proxy for mitochondrial $Ca^{2+}$ uptake, was increased following MICE only when combined with OLE, along with a stronger mitochondrial transcriptional response independent of changes in mitochondrial respiratory complex protein levels. Conversely, OLE did not further increase PDH activity in response to SIE but also increased

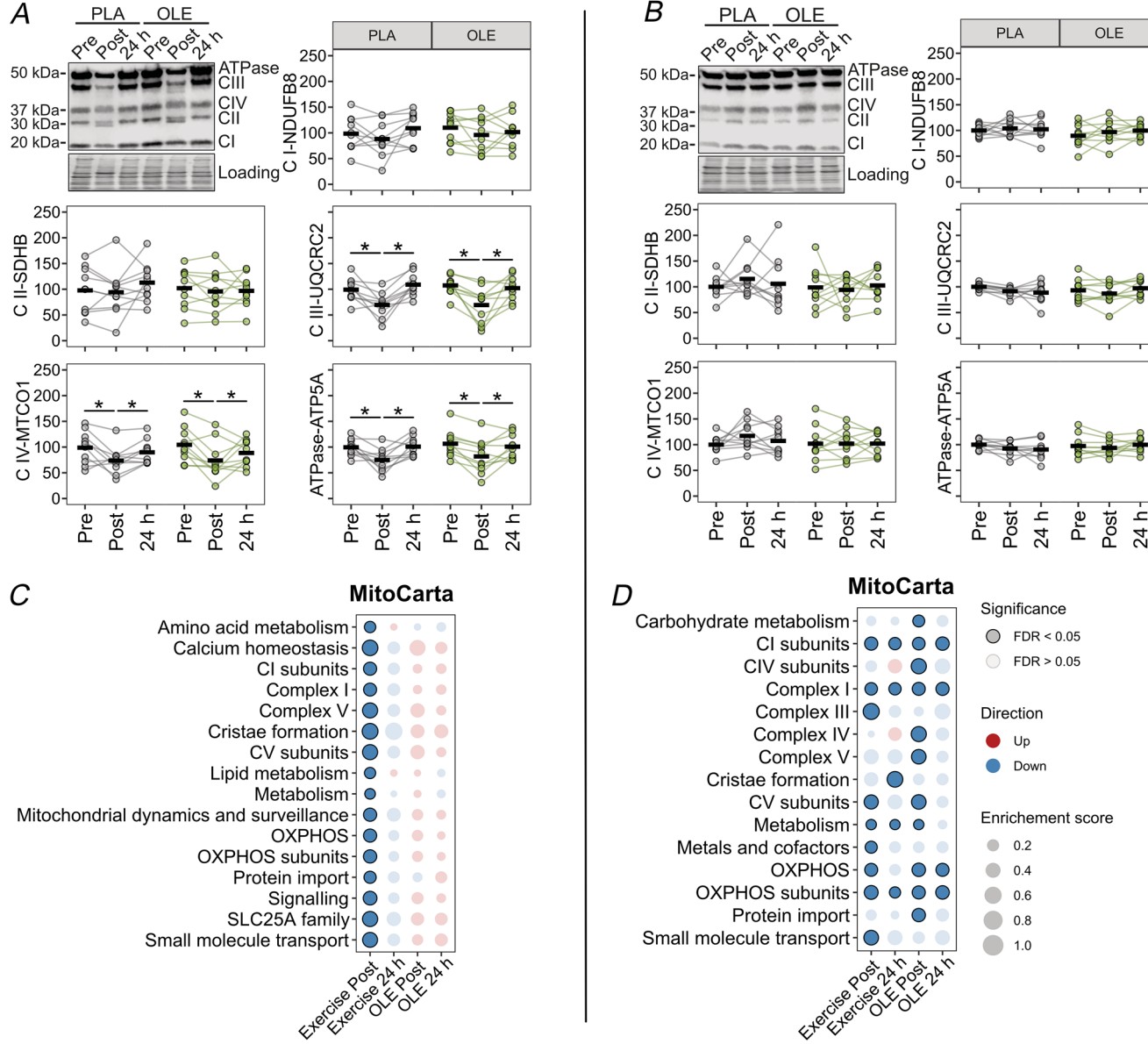

**Figure 8. Olive leaf extract (OLE) effect on mitochondrial-related proteome following exercise**

*A* and *B*, representative immunoblots of complex I, II, III, IV and ATPase subunits and their respective quantification. Measurements were performed on samples collected before (Pre), immediately after (Post) and 24 h after (24 h) a session of MICE (*A*) or SIE (*B*) with ingestion of placebo (PLA) or olive leaf extract (OLE). Black thick lines represent the mean and dots linked with lines represent individual participants. *$P < 0.05$. *C* and *D*, dot plots representing the enrichment of proteomic pathways from the MitoCarta database following MICE (*C*) and SIE (*D*) with ingestion of PLA or OLE. Columns represent the effect of exercise + PLA at Post (Exercise Post) and at 24 h (Exercise 24 h) and effect of OLE corrected for the effect of exercise at Post (OLE post) and at 24 h (OLE 24 h). For all panels, data were obtained from 10 participants.

the mitochondrial transcriptional response. This suggests that the milder or longer mitochondrial stimulation from MICE may be more permissive to a synergy between OLE and exercise on mitochondrial $Ca^{2+}$ uptake and its downstream response. OLE may also act through alternative pathways than PDH activation as OLE intake improved the early resistance to fatigue and altered the mitochondrial response during SIE, which were not associated with any additional changes in PDH activity. Furthermore, regulation of $Ca^{2+}$ homeostasis via mitochondrial $Ca^{2+}$ can modulate cell metabolism, cell survival and other cell-type-specific functions, in addition to PDH activation (Rizzuto et al., 2012). The exercise protocols used in this study were primarily designed to test acute exercise adaptation and were not optimized for performance evaluation since exercise duration and intensity were fixed per protocol. Future studies will therefore be important to evaluate the impact of OLE on acute exercise performance as well as its effect on chronic training.

The transcriptional response to exercise has been extensively studied in the past (Amar et al., 2021; Edman et al., 2024; Pillon et al., 2020). Recent meta-analyses indicated a relatively low number of genes being regulated following one session of exercise as compared to our results (Amar et al., 2021; Pillon et al., 2020). As

suggested by recent research that reported a comparable or even greater number of regulated genes following MICE and SIE relative to our findings (Botella et al., 2024), previous studies may have underestimated the transcriptional impact of exercise by neglecting its dynamic and prolonged temporal response, thereby overlooking the regulation of certain genes. Indeed, the number of regulated genes 24 h following exercise was comparable to that observed immediately after exercise but with only a small overlap between the regulated genes at both time points. The relatively low number of regulated genes immediately following SIE as compared to MICE was surprising but may result from the very short period between the start of exercise and the following biopsy (23 min for SIE *vs.* 60 min for MICE), limiting the time for the transcriptional machinery to trigger detectable changes. Interestingly, a very small proportion of the regulated genes were shared between the two time points, showing the time-specific response of transcription following exercise (Egan & Sharples, 2023). This is further highlighted by the clustering analysis, showing a variety of dynamics between different groups of genes. Overall, our results stress the importance of carefully choosing the time points when investigating transcriptional responses to exercise based on specific mechanisms of interest.

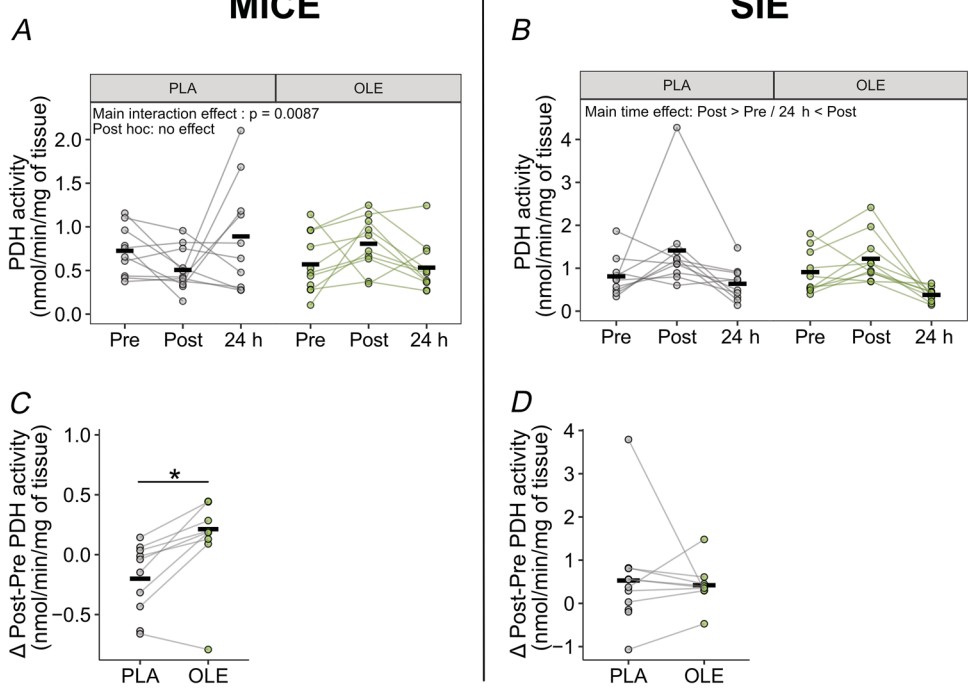

**Figure 9. Effect of exercise combined with olive leaf extract (OLE) on PDH activity**
*A* and *B*, PDH activity measured in samples collected before (Pre), immediately after (Post) and 24 h after (24 h) a session of MICE (*A*) or SIE (*B*) combined with placebo (PLA) or OLE ingestion. *C* and *D*, delta changes from Pre to Post in response to MICE (*C*) or SIE (*D*) with ingestion of PLA or OLE. All time points were obtained from nine participants. Black thick lines represent the mean and dots linked with lines represent individual participants. *$P \leq 0.05$.

We also investigated whether acute exercise might cause changes at the proteome level. As expected, the top upregulated protein pathways following MICE and SIE were mainly associated with immune/inflammatory processes (Peake et al., 2017). Following 24 h of recovery, the proteome was almost back to its original state with only a few proteins still affected after MICE. Interestingly, both targeted investigations and proteomics showed a transient but consistent down-regulation of mitochondrial-related proteins immediately after exercise. Following SIE, this reduction was supported by a concomitant decrease in mitochondria-related mRNA. This was not the case following MICE, suggesting that other processes such as mitophagy may have contributed to this phenotype (Ritenis et al., 2025). Interestingly, this result may be consistent with recent evidence indicating that acute exercise can trigger mitophagy and the release of mitochondria from muscle fibres (Díaz-Castro et al., 2024). Contrasting with previous results showing an increase in protein content of complexes I, II and IV of the electron transport system 24 h after SIE and of complex IV 24 h after MICE (Zanou et al., 2021), we did not observe any increase of OXPHOS proteins at 24 h in the present study. This result indicates that multiple exercise sessions may be needed to induce a consistent increase in mitochondrial protein level (Egan & Sharples, 2023).

To determine whether OLE influences the muscle response to exercise, we examined its effect on PDH activity. In accordance with our hypothesis, the activation of PDH after MICE and OLE ingestion suggests an increase in mitochondrial $Ca^{2+}$ uptake and a possible link with the mitochondrial bioenergetic adaptations (Gherardi et al., 2020). Interestingly, OLE intake markedly enhanced the mitochondrial transcriptional response 24 h after MICE, affecting a broad range of mitochondrial pathways. MICE is known to activate mitochondrial transcription factors (Pillon et al., 2020), and our results suggest that this activation is further amplified in the presence of OLE. A direct link between the potentiation of the transcriptional response by OLE and the increase in PDH activity cannot be shown in the present study. However, acute OLE intervention in preclinical studies with mice increased PDH activity and chronic ingestion feeding enhanced mitochondrial respiration and aerobic performance, while this effect was lost in MCU knock-out mice (Gherardi et al., 2025). In addition, previous work highlighted the importance of mitochondrial $Ca^{2+}$ uptake during exercise to drive mitochondrial adaptations to exercise (Zanou et al., 2021). This suggests that OLE intake potentiated the mitochondrial response to MICE, at least in part, through an increase in mitochondrial $Ca^{2+}$ uptake and bioenergetics. In addition, a recent human study in healthy older males observed a 25% increase in PDH activity following chronic supplementation, but not 2 h after OLE supplementation at rest (Pinckaers et al., 2025). Hence, muscle contraction may be important to allow OLE to increase mitochondrial $Ca^{2+}$ uptake and PDH activity, probably by inducing a large release of $Ca^{2+}$ from the sarcoplasmic reticulum to the cytosol.

PDH activity in response to SIE was already strongly increased following exercise alone and OLE did not induce any additional effects. It is possible that SIE, as a very intense exercise modality, is already inducing maximal activation of PDH (Putman et al., 1995) and that OLE intake is therefore unable to further increase this activation. SIE alone induced a general down-regulation of mitochondrial pathways immediately after exercise, with OXPHOS genes remaining low 24 h later. However, the genes involved in mitochondrial replication appeared up-regulated. These findings suggest that 24 h following SIE may be too early to observe a full recovery or a transcriptional up-regulation of OXPHOS genes, suggesting that the muscle fibres may still be responding to the initial stressing effect of exercise. Surprisingly, the transcriptional effects of OLE appear to counter those of exercise. Immediately after SIE, OLE mitigated the down-regulation of OXPHOS genes, and by 24 h, it strongly up-regulated mitochondrial gene expression as compared to exercise alone. One possible explanation is that OLE, by blunting the initial suppressive response triggered by exercise, accelerates the onset of the mitochondrial transcriptional initiation. Unfortunately, the absence of a later time point for analysis does not allow us to clarify this possibility. Since PDH activity was unaffected by OLE following SIE, the observed effect of OLE on mitochondrial transcription suggests either that low-amplitude modulation of mitochondrial $Ca^{2+}$ uptake is too mild to induce detectable PDH activation on top of the strong effects of SIE, or that OLE modulates the mitochondrial response to SIE through alternative mechanisms.

To identify potential effects of OLE beyond increased mitochondrial $Ca^{2+}$ uptake, we annotated our transcriptomic data using the broader HALLMARK gene set database. The most striking changes were observed in pathways related to the inflammatory response. The immediate potentiation of the transcriptional inflammatory response after MICE with OLE may indicate an increase in the immune system recruitment classically observed in the early recovery phase of exercise (Peake et al., 2017). The absence of an OLE effect after SIE may be explained by the different time intervals between the start of exercise and the Post biopsy for the two exercise modalities, as previously highlighted in the Discussion. Interestingly, the prolonged transcriptional inflammatory response 24 h after SIE appears to be strongly dampened by OLE intake. Olive-derived

polyphenols have been shown to reduce markers of the inflammatory response following exercise in rodents and in human (Lillis et al., 2025). In addition, oleuropein has been shown to reduce inflammation *in vitro*, by acting on macrophage polarization (Mirsanei et al., 2023). The physiological inflammatory response following exercise is important for muscle mitochondrial adaptations after training (Langston & Mathis, 2024; Wyckelsma et al., 2026). Therefore, although the increased mitochondrial $Ca^{2+}$ uptake expected with OLE intervention probably contributes to the mitochondrial response following MICE, other targets of OLE involved in the inflammatory process may modulate this response – particularly after SIE, where the inflammatory reaction is stronger due to the higher intensity (Peake et al., 2017). Therefore, supplementation of anti-inflammatory nutrients during training needs to be carefully assessed, especially for high-intensity modalities inducing large inflammatory responses where a setpoint exists between pathological hyper-inflammation and the physiological inflammatory response that triggers beneficial adaptations (Merry & Ristow, 2016; Wyckelsma et al., 2026).

During MICE and SIE, HR tended to be lower when participants ingested OLE, although these modest differences did not reach statistical significance and may have been constrained by the sample size. Previous studies in rodent models have shown that oleuropein acutely reduces HR at rest (Ilic et al., 2021; Ivanov et al., 2018). While the exact mechanisms remain unclear, it is unlikely to arise from a direct effect of OLE on heart calcium/contraction pairing as we did not detect any effect of OLE at its active concentration on primary human cardiomyocyte calcium release or contractility (Nestlé Research, data not shown). The potential influence of OLE on cardiac features at rest and during exercise should be more closely assessed in future adequately powered studies.

The lower fatigue index observed during the first sprint of the SIE session when ingesting OLE confirmed recent data in mice indicating a better resistance to fatigue in the early stage of a repeated tetanic contraction protocol with OLE ingestion (Gherardi et al., 2025). While a single 30 s all-out sprint (Wingate test) is usually used to test anaerobic capacity, it has been shown that aerobic metabolism contributes to around 15–20% of the work performed (Smith & Hill, 1991), which may explain how OLE improved the fatigue index. Normalization of the phenotype in the later stages of the SIE could be explained by the onset of sub-stantial peripheral fatigue as expected from repeated high-intensity exercise, which may have hindered the effects of OLE. Indeed, an important overall fatigability was observed following both exercise modalities, mainly explained by peripheral alterations as suggested by the

reduction in PD force. The preservation of M-wave amplitude, combined with a decreased 10/100 Hz ratio, suggests that these alterations are unlikely to result from impaired sarcolemmal excitability, but rather from impaired $Ca^{2+}$ handling (Zanou et al., 2021). No effect of OLE intake was observed on the extent of peripheral and $Ca^{2+}$ handling alterations following MICE and SIE in the present study. This suggests that mitochondrial $Ca^{2+}$ uptake does not modify cytosolic $Ca^{2+}$ handling and is not a major contributor of peripheral fatigue mechanisms after prolonged exercise. Alternatively, the potential OLE-mediated stimulation of mitochondrial $Ca^{2+}$ uptake may be insufficient in magnitude or duration to mitigate the fatigability phenotype beyond its early effects during the first high-intensity contractions.

While the present study provides valuable insights into the acute mitochondrial response of OLE supplementation in combination with acute exercise, caution should be taken when interpreting the results due to several limitations. First, no direct measurement of mitochondrial $Ca^{2+}$ uptake was performed because of the complexity of this assessment in human muscle and clinical trials. While mitochondrial $Ca^{2+}$ uptake is a strong mediator of PDH activity during exercise, other metabolic parameters could also play a role. Second, the inflammatory modulation by OLE is inferred solely from RNA-seq pathway analysis, as no direct measurements of cytokines or activation of inflammatory signalling pathways were performed. Since mRNA changes do not always translate into actual cytokine abundance or signalling activity alterations, future studies should directly characterize inflammatory markers in muscle when investigating the effects of OLE on inflammation. As highlighted by our own analysis, the transcriptional response to exercise is very dynamic and depends on the chosen exercise modality and kinetics analysed. The moments at which biopsies were performed in the present study were the same for both MICE and SIE, and may not best adapt to the individual kinetics of the response to each exercise type. We also acknowledge that post-intervention measurements of biopsy-related parameters, particularly those involving immune responses, may have been partially influenced by tissue injury caused by the pre-intervention biopsy. However, while effects of repeated biopsy sampling on inflammatory markers have been documented when biopsies were separated by 48 h (Van Thienen et al., 2014), evidence suggests that such effects are limited within the time frame relevant to our study (30–60 min) (Guerra et al., 2011). Finally, this study investigated the acute response to a single bout of exercise, and it is likely that both chronic exercise and OLE consumption will trigger further adaptations.

In conclusion, the present study showed that OLE intake combined with MICE potentiates the

mitochondrial response after exercise, via a mechanism that potentially involves mitochondrial $Ca^{2+}$ uptake and PDH activation. Our results also suggest that OLE might affect other mechanisms such as inflammatory mediators. The performance improvement observed during SIE and HR adaptation during MICE may result from these muscle-specific alterations as well as from more systemic effects of OLE. The long-term effect of OLE intake in combination with exercise will need to be assessed in chronic training studies specifically designed from the conclusions of this work. Together, this study highlights the potential of OLE to improve the mitochondrial response following exercise, which may be beneficial for both healthy individuals and populations at risk of pathological decline.

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

## Additional information

### Data availability statement

The mass spectrometry proteomics data have been deposited at the ProteomeXchange Consortium via the PRIDE (Perez-Riverol et al., 2025) partner repository with the dataset identifier PXD072766. RNA sequencing data have been deposited with the GEO expression omnibus with the dataset identifier GSE318937. These data will be made publicly accessible. All other data supporting the findings of the present study are available from the corresponding author upon reasonable request. Access will be granted in compliance with ethical and privacy regulations and only for purposes that align with the study aims and participants' consent.

### Competing interests

A.M.-A., A.M.H.H., E.M., O.C., J.S., S.M., A.H., A.L.P., L.D., U.D.M. and J.N.F. are employees of Société des Produits Nestlé

(SPN) SA. A.H., U.D.M. and J.N.F. are inventors on patents owned by SPN SA on the use of oleuropein and olive leaf extract for muscle health and physical energy. Other authors declare no conflicts of interest.

## Author contributions

C.L., A.M.H.H., J.N.F, U.D.M., N.Z. and N.P. designed the research. C.L., S.M., E.M., O.C., J.S., S.M., A.M.H.H., A.L.P, L.D., N.Z. and N.P. participated in the data collection and/or the data and sample analyses. C.L., S.M., A.M.-A., A.H., A.M.H.H., U.D.M., J.N.F., N.Z. and N.P. were involved in data interpretation. C.L. and S.M. drafted the manuscript and all other authors revised the manuscript. All authors have approved the final version of the manuscript submitted for publication and agree to be accountable for all aspects of the work. All people designated as authors qualify for authorship, and all those who qualify for authorship are listed.

## Funding

The clinical trial was sponsored by the University of Lausanne.

## Acknowledgements

Nestlé Research provided the study product and performed in-kind transcriptomic and proteomic analyses.

Open access publishing facilitated by Université de Lausanne, as part of the Wiley - Université de Lausanne agreement via the Consortium Of Swiss Academic Libraries.

## Author's present address

A. M. H. Horstman: Faculty of Health, Nutrition and Sport, The Hague University of Applied Sciences, The Hague, the Netherlands.

## Keywords

calcium mitochondria, muscle fatigue, oxidative phosphorylation, power output, pyruvate dehydrogenase

## Supporting information

Additional supporting information can be found online in the Supporting Information section at the end of the HTML view of the article. Supporting information files available:

**Peer Review History**

