## [Peer Review History · The Journal of Physiology]

Oleuropein-based olive leaf extract enhances muscle mitochondrial bioenergetics response to moderate –but not maximal- intensity exercise in humans

Clément Lanfranchi, Alba Moreno-Asso, Astrid MH Horstman, Sara Mistro, Eugenia Migliavacca, Ornella Cominetti, Jens Stolte, Sylviane Métairon, Aurélie Hermant, Ane Laura Fineid Pedersen, Loïc Dayon, Umberto De Marchi, Jerome N Feige, Nadège Zanou, and Nicolas Place

DOI: 10.1113/JP290316

Corresponding author(s): Nicolas Place (nicolas.place@unil.ch)

Review Timeline:

Submission Date:	13-Oct-2025
Editorial Decision:	24-Nov-2025
Revision Received:	05-Jan-2026
Editorial Decision:	09-Feb-2026
Revision Received:	27-Feb-2026
Accepted:	19-Mar-2026

Senior Editor: Paul Greenhaff

Reviewing Editor: Jørn Helge

Transaction Report:

Re: JP-RP-2025-290316 "An Oleuropein-based olive leaf extract enhances the muscle mitochondrial response to acute moderate - but not maximal - intensity exercise in humans" by Clément Lanfranchi, Alba Moreno-Asso, Astrid M Horstman, Sara Mistro, Eugenia Migliavacca, Ornella Cominetti, Jens Stolte, Sylviane Métairon, Aurélie Hermant, Ane Laura Pedersen, Loïc Dayon, Umberto De Marchi, Jerome Feije, Nadège Zanou, and Nicolas Place

Dear Dr Place,

Thank you for submitting your manuscript to The Journal of Physiology. It has been assessed by a Reviewing Editor and by 2 expert referees and we are pleased to tell you that it is potentially acceptable for publication following satisfactory major revision.

Please address all the points raised and incorporate all requested revisions or explain in your Response to Referees why a change has not been made. We hope you will find the comments helpful and that you will be able to return your revised manuscript within 2 months. If your article is NOT for a Special Issue, you may have 9 months to revise. If you require an extension, please contact journal staff: jp@physoc.org. Please note that this letter does not constitute a guarantee for acceptance of your revised manuscript.

REVISION CHECKLIST:

Upload a full Response to Referees file. To create your 'Response to Referees': copy all the reports, including any comments from the Senior and Reviewing Editors, into a Microsoft Word, or similar, file and respond to each point, using

font or background colour to distinguish comments and responses and upload as the required file type.

We look forward to receiving your revised submission.

Yours sincerely,

Paul Greenhaff
Senior Editor
The Journal of Physiology

REQUIRED ITEMS

- The Journal of Physiology funds authors of provisionally accepted papers to use the premium BioRender site to create high resolution schematic figures. Follow this link and enter your details and the manuscript number to create and download figures. Upload these as the figure files for your revised submission. If you choose not to take up this offer, we require figures to be of similar quality and resolution. If you are opting out of this service to authors, state this in the Comments section on the Detailed Information page of the submission form. The link provided should only be used for the purposes of this submission. Authors will be charged for figures created on this premium BioRender account if they are not related to this manuscript submission.

- Please upload separate high-quality figure files via the submission form.

- You must upload original, uncropped western blot/gel images (including controls) if they are not included in the manuscript. This is to confirm that no inappropriate, unethical or misleading image manipulation has occurred. These should be uploaded as 'Supporting information for review process only'. Please label/highlight the original gels so that we can clearly see which sections/lanes have been used in the manuscript figures. For more information, see: <https://physoc.onlinelibrary.wiley.com/hub/journal-policies#imagmanip>.

EDITOR COMMENTS

Reviewing Editor:

Comments to the Author:

Authors have studied the influence of OLE, novel for human application, on mitochondrial adaptations following two exercise modes. The paper is interesting and relevant. The two reviewers are positive, but have a number of questions, comments and clarifications that must be addressed.

Senior Editor:

Comments to the Author:

Thank you for the manuscript submission to The Journal of Physiology, which has been considered by a reviewing editor and two reviewers. The topic is of interest to the Journal readership and is viewed to have been well written. However, a number of concerns have been received that the authors must address adequately if the manuscript is to be considered further. In particular, the point of Reviewer 1 that it is questionable the measurement of PDH activity can be viewed as a proxy for mitochondrial calcium uptake appears to be a fundamental point. As flagged by Reviewer 1, the homogenisation buffer contained 5 mM EGTA, which is known to buffer calcium. Secondly, it also contained dichloroacetate, which is a

known PDK inhibitor that activates PDH in a non-calcium dependent manner. Both these points need consideration. Finally, given PDH activation state during exercise correlates to the intensity of exercise performed in human volunteers [<https://doi.org/10.1111/j.1748-1716.1991.tb09247.x>], and that there is inertia in PDH activation at the onset of exercise [DOI: 10.1111/j..2002.t01-1-00591.x], the authors need to consider the impact of exercise intensity and duration on their reported findings when comparing sprint interval exercise and moderate-intensity continuous exercise.

REFeree COMMENTS

Referee #1:

Here the authors investigated whether oleuropein (OLE), either alone or combined with two types of exercise, enhances mitochondrial Ca²⁺ uptake and thereby promotes mitochondrial adaptations. This is based on previous studies in mice and cultured cells, as well as the role of mitochondrial Ca²⁺ uptake for mitochondrial adaptation to exercise. PDH activity was used as an indicator of mitochondrial Ca²⁺ uptake, while mitochondrial adaptations were assessed through mitochondrial transcriptional responses and respiratory complex protein levels. The acute exercise modalities were sprint interval exercise (SIE) and moderate-intensity continuous exercise (MICE). These were chosen because SIE has previously been shown to trigger mitochondrial Ca²⁺ uptake, whereas the effects of MICE are less clear.

The fatigue index across the six SIE and the first sprint demonstrated a reduction with OLE only in the first sprint. While both exercise modalities impaired muscle force generating capacity, OLE did not modify this response.

PDH activity increased only after SIE, but not MICE. In contrast OLE, in combined with exercise, increased PDH activity in MICE, but did not further increase it after SIE. Both exercise and OLE independently led to a stronger mitochondrial transcriptional response, which occurred without corresponding changes in respiratory complex protein levels. Interestingly, while OLE did not further elevate PDH activity following SIE, it still augmented the mitochondrial transcriptional response.

The authors are commended for conducting the first human study examining the effects of oleuropein (OLE) on exercise, translating preclinical findings into a human context by investigating how an oleuropein-based olive leaf extract influences muscle mitochondrial responses to different exercise types. However, I have several concerns and comments regarding the study's findings and interpretations.

Specific comments

OLE. Oleuropein is a major phenolic compound found in olive leaves and unprocessed olives, but it's mostly absent from finished olive oil because it is transformed or removed during processing. Still there is a substantial amount, in particular green olive fruits, with some claims of nearly same amount per wet weight. Inclusion criteria was "... stable eating habits" (line 112). Did you control for the subject's olive intake (green!) during the test/study?

Hypothesis. PDH activity as a proxy for mitochondrial Ca²⁺ uptake, and do not measure mitochondrial Ca²⁺ contents (which admittedly is difficult). yet, you state that the purpose is to test if OLE "...is triggering mitochondrial Ca²⁺ uptake and PDH activation during MICE. You only measure PDF activity and not mitochondrial Ca²⁺ contents, thus this is not valid to state both.

Biopsies. The central conclusions about OLEs modulation of inflammation and enhancement of mitochondrial pathways rely on bulk RNA-seq and proteomics immediately post-exercise and at 24 h (Figures 5, 6, and 7). Pre and Post biopsies were collected through the same incision, with multiple samples from same puncture site. By using the same incision for both Pre and Post muscle biopsies could introduce local injury/inflammation that confounds signals attributed to exercise or OLE (Figs 5-7). Thus, repeated needle biopsies cause disturbances in myocellular signaling pathways (doi:10.14814/phy2.286). I'm concerned if the acute immune pathway enrichments and OLE effects are affected by biopsy-induced tissue injury and thereby the interpretation of acute inflammatory and mitochondrial pathway changes. Please comment on this possibility and justify.

Mitochondrial effects. The proteomic and targeted analyses revealed a downregulation of mitochondrial-related pathways immediately after MICE while only proteomics indicated a downregulation after SIE. Decreases were observed for Complex III, IV, and V subunits after MICE, while no changes after SIE. Further, the mitochondrial pathway downregulation was reasoned from proteomic enrichment. As the protein analysis was assessing only one subunit per OXPHOS complex and without normalization to mitochondrial content markers or broader mitochondrial protein panels, limits the generalization to

pathway-level downregulation and the assertion that only proteomics detected changes after sprint interval exercise. Studies on effects of exercise on OXPHOS subunits show a non-stoichiometric change (mitochondrial contents / analysed) in response to acute exercise, and most training-induced changes in mitochondrial functional groups become non-significant when normalized to mitochondrial content markers as e.g. CS. As the data presented are a narrow set of proteins estimates of mitochondrial abundance, the observed decreases and the lack of changes may not represent widespread pathway behaviour. Please justify the claims.

PDH activity. PDH activity was increased following MICE only when combined with OLE, while OLE did not further increase PDH activity in response to SIE. The PDH activity in the homogenized biopsy material was estimated using a standard assay kit. The homogenisation buffer contained 5 mM EGTA and 5 mM dichloroacetate (DCA), which buffer Ca²⁺ (EGTA) and is known to inhibit PDK (DCA). Thus, I fear that the assumed Ca²⁺ dependent activation of PDH, i.e. preventing the Ca²⁺ dependent dephosphorylation of PDH (e.g. doi: 10.1042/bj2180235), is confounded and interpretation of PDH activation as a proxy for mitochondrial Ca²⁺ uptake is not valid. Further, DCA is known to inhibit PDK thereby activating PDH in a non-Ca²⁺ depending way, (e.g. doi.org/10.1210/jc.2005-0123). Together, the observed increase in PDF activity can be attributed to (or possibly are) homogenising buffer, weakening the mechanistic link central to the claim. Together, the assays need to be performed in Ca²⁺ controlled and DCA-free conditions to verify the assay validity and hence mechanistic claim.

Mitochondrial Ca²⁺ buffering. Lines 629-631 and throughout the manuscript its stated that mitochondrial Ca²⁺ buffering can modulate cell metabolism. I understand its debated and mitochondrial Ca²⁺ uptake activates metabolic activity, however, as there are no apparent Ca²⁺ buffers in the mitochondria the Ca²⁺ buffering capacity is relatively limited. Please discuss the issue of mitochondrial buffering and quantitative importance for cellular Ca²⁺ regulation.

Fatigue index. Lines 435-440. The fatigue index across six sprints and the first sprint, showing a significant reduction with the supplement only in the first sprint. The improvement in early-stage fatigue resistance is inferred from a lower fatigue index in the first 30-second sprint with olive leaf extract (Fig 2E), alongside the progression of fatigue index across sprints (Fig 2D) and the comparison of first sprint mean power (Fig 2C). The analysis does not rule out that the lower fatigue index in the first sprint reflects differences in initial power or acceleration rather than true resistance to fatigue. The fatigue index depends on the change between the first and last 5 seconds of the sprint (Fig 2E), and differences in early power profile will affect this index. Were there any differences in initial power between trials?

Heart rate (p. 15). The stated reduction in mean heart rate during moderate cycling does not meet statistical significance, i.e. p-value 0.0547 (Fig 2A). However, there is definitely a trend-level which likely may be constrained by sample size. Please rewrite and modify the claim to reflect this.

Inflammatory signalling. The overall conclusion that OLE enhances acute inflammatory signalling after MICE and attenuates it after SIE at 24 h is based on pathway enrichment analysis of bulk muscle RNA-seq data. Measures of "HALLMARK pathways" (Fig 6A-B) exhibit consistent transcript changes, and proteomics data (Figures 5C-D) confirm that exercise induces immune and inflammatory responses. This is carefully generated. Still, inflammatory modulation is concluded from RNA-seq pathways without direct protein-level or cellular validation. Ill suggest that the authors clearly state and discuss that the mRNA changes may not reflect actual cytokine abundance or signalling activity, as well as the proteomics analysis does not specifically assess OLE-related inflammatory effects.

Also, the acute comparison between modalities is confounded by sampling times for the Post biopsy sampling time relative to exercise start and OLE administration. This may affect or bias the detection of immediate effects after SIE. The observed lack of OLE effect after SIE could reflect insufficient time for transcript, which is observed in MICE where biopsy is taken noticeably later leaving changes to manifest. Please discuss that in context to the overall conclusion that OLE enhances acute inflammatory signalling after MICE and attenuates it after SIE.

Assessments. You very precisely assessed both neuromuscular parameters and torque and EMG during knee extension. Its not clear by the reader why and relevance of these measures for the overall aim of test the hypothesis that " .. combining OLE to SIE or MICE may influence muscle bioenergetics .. ". I may have missed it, but please explain the purpose using these estimates and possible interpretations of the results. Also, lines 225-229, please explain briefly the assumption that PS10/100 is an index of intracellular Ca²⁺ handling. Do you mean release?

Title. Please be more specific in title on "mitochondrial response" its unclear to the reader what meant. I understand that it's a relatively broad range of mitochondrial affects you are determining, but consider using more specific term(s), i.e. mitochondrial transcriptional response and respiratory complex protein levels.

Minor

Lines 68-71. Off note, the leaky RyR1 was observed only in untrained subjects/unaccustomed exercise. While trained still have "beneficial metabolic adaptations" with no apparent RyR1 leak. Please state clearly that training adaptations is only partly explained by the Ca²⁺ Leak/release effects.

Line 79. Therefore, we wondered State clearly if this is a hypothesis. Likewise, line 101, ... we reasoned I believe you hypothesize.

Introduction. Lines 61.62. Tha justification of SIE training as being more "time efficient" is a common accepted assertion. A quick calculation reveals that your MICE protocol last just more than 1 hr and the SIE, all in all 30 min. It would be more relevant to state the superior effects on circulation and health parameters with HIIT/SIE compared to MICE, due to also activation of fast twitch fibres.

Figure 9. The increased PDH activity post SIE in PLA (Fig 9B) seems driven by one person. Are there any effects omitting the subject?

Line 595. "Since OLE activates bioenergetics by stimulating PDH activity via increased mitochondrial Ca²⁺ uptake during contraction, Please add reference.

Line 120. Did you estimate VO₂ during the incremental cycling protocol. i.e. VO₂max? and how was MAP defined, i.e. last

Line 435. Power decrease, give P-value.

Line 437. How is "anaerobic performance" defied.

Referee #2:

I have been reading the paper by Lanfranchi and colleagues, where they are looking into different training modalities effect on mitochondrial bioenergetics, mitochondrial calcium uptake and pyruvate dehydrogenase activation. Furthermore they also looked into the compound oleuropeins effect on these parameters mentioned before. The study was conducted as a double-blinded cross-over study in healthy males. Muscle biopsies were obtained before, immediately after and 24 hours after an acute exercise bout (moderate intensity: 1 h at 50% maximal aerobic power or sprint interval exercise 6 × 30 s all-out sprints with 4 min recovery). I have some comments to the manuscript.

Comments:

Line 73-75: The study by Zanou et al. was conducted in C2C12 cells as I remember, maybe add this information.

Line 77: What clinical populations are unable to perform intense exercise, please clarify that and add a reference.

Line 110: The introduction are focusing a lot of health burden and chronic diseases, why then investigate it in young healthy men and why only men?

Line 112: What is the meaning of good health and stable eating habits, please clarify that.

Line 114: Should it not be "performed at..."

Line 128: Was the taste of the product and the placebo the same?

Line 135: Why was this timepoint (30 min after ingestion) picked?

Line 161-164: Has the Authors thought about if there is any issues with their analysis and the sampling of muscle biopsies? Could this in any way affect their outcome? Please add this in the revised manuscript.

Line 195: What was the Authors experience with compensatory movements?

Line 205: Was the two exercise protocols matched for energy used?

Line 264: How long time after the muscle biopsies was taken was the analysis done? How long was the samples stored in the freezer?

Line 418: Please clarify what several issues were?

Line 508: Very interesting, where the biopsies in the present study taken at the same time as the study by Layec, 2018?

Line 524-525: What is meant by slightly lower?

Line 567-568: It is surprising for me that the protein levels changes that fast, could the Authors please elaborate on that?

Line 594: As far as I can see calcium uptake was not measured in the present study, why not?

END OF COMMENTS

EDITOR COMMENTS

Reviewing Editor:

Comments to the Author:

Authors have studied the influence of OLE, novel for human application, on mitochondrial adaptations following two exercise modes. The paper is interesting and relevant. The two reviewers are positive, but have a number of questions, comments and clarifications that must be addressed.

We thank the reviewing editor for the positive evaluation of our manuscript. The complete answers to the Senior Editor and the two reviewers are provided below.

Senior Editor:

Comments to the Author:

Thank you for the manuscript submission to The Journal of Physiology, which has been considered by a reviewing editor and two reviewers. The topic is of interest to the Journal readership and is viewed to have been well written. However, a number of concerns have been received that the authors must address adequately if the manuscript is to be considered further. In particular, the point of Reviewer 1 that it is questionable the measurement of PDH activity can be viewed as a proxy for mitochondrial calcium uptake appears to be a fundamental point. As flagged by Reviewer 1, the homogenisation buffer contained 5 mM EGTA, which is known to buffer calcium. Secondly, it also contained dichloroacetate, which is a known PDK inhibitor that activates PDH in a non-calcium dependent manner. Both these points need consideration. Finally, given PDH activation state during exercise correlates to the intensity of exercise performed in human volunteers [<https://doi.org/10.1111/j.1748-1716.1991.tb09247.x>], and that there is inertia in PDH activation at the onset of exercise [DOI: 10.1111/j..2002.t01-1-00591.x], the authors need to consider the impact of exercise intensity and duration on their reported findings when comparing sprint interval exercise and moderate-intensity continuous exercise.

First, we want to thank the Senior Editor for the interest regarding our manuscript.

We have indeed answered the point raised by Reviewer 1 concerning PDH activity and the homogenisation buffer below. In brief, these various chemicals aim at blocking any reactions that could occur after the muscle biopsy sampling, thereby ensuring that PDH activity is measured under conditions in which its *in vivo* phosphorylation state is preserved.

Regarding the impact of exercise intensity and duration on PDH activity, it is indeed well established that PDH activation is intensity dependent. In line with this, our findings indicate that SIE combined with placebo markedly activated PDH, whereas MICE did not. We already stated in the discussion that it is likely that "SIE, as a very intense exercise modality, is already inducing maximal activation of PDH (Putman et al., 1995) and that OLE intake is therefore unable to further increase this activation." In addition, while the primary aim was not to directly compare the SIE and MICE group, we agree that the Post biopsy timing of SIE and MICE (~30min and ~60min after the onset of exercise respectively) may explain some of the observed differences beyond the sole effect of exercise modality. This consideration is discussed in the limitations section: "As highlighted by our own analysis, the transcriptional response to exercise is very dynamic and depends on the chosen exercise modality and kinetics analyzed. The moments to which biopsies were performed in the present study were the same for both MICE and SIE, and may not best adapt to the individual kinetic of the response to each exercise type."

These points are discussed in the specific answers provided below.

REFEREE COMMENTS

Referee #1:

Here the authors investigated whether oleuropein (OLE), either alone or combined with two types of exercise, enhances mitochondrial Ca^{2+} uptake and thereby promotes mitochondrial adaptations. This is based on previous studies in mice and cultured cells, as well as the role of mitochondrial Ca^{2+} uptake for mitochondrial adaptation to exercise. PDH activity was used as an indicator of mitochondrial Ca^{2+} uptake, while mitochondrial adaptations were assessed through mitochondrial transcriptional responses and respiratory complex protein levels. The acute exercise modalities were sprint interval exercise (SIE) and moderate-intensity continuous exercise (MICE). These were chosen because SIE has previously been shown to trigger mitochondrial Ca^{2+} uptake, whereas the effects of MICE are less clear.

The fatigue index across the six SIE and the first sprint demonstrated a reduction with OLE only in the first sprint. While both exercise modalities impaired muscle force generating capacity, OLE did not modify this response.

PDH activity increased only after SIE, but not MICE. In contrast OLE, in combined with exercise, increased PDH activity in MICE, but did not further increase it after SIE. Both exercise and OLE independently led to a stronger mitochondrial transcriptional response, which occurred without corresponding changes in respiratory complex protein levels. Interestingly, while OLE did not further elevate PDH activity following SIE, it still augmented the mitochondrial transcriptional response.

The authors are commended for conducting the first human study examining the effects of oleuropein (OLE) on exercise, translating preclinical findings into a human context by investigating how an oleuropein-based olive leaf extract influences muscle mitochondrial responses to different exercise types. However, I have several concerns and comments regarding the study's findings and interpretations.

We thank the reviewer for acknowledging the innovative study we conducted.

Specific comments

OLE. Oleuropein is a major phenolic compound found in olive leaves and unprocessed olives, but it's mostly absent from finished olive oil because it is transformed or removed during processing. Still there is a substantial amount, in particular green olive fruits, with some claims of nearly same amount per wet weight. Inclusion criteria was "... stable eating habits" (line 112). Did you control for the subject's olive intake (green!) during the test/study?

For both experimental sessions, participants were asked to eat the same last meal before each session and were asked to report it to the investigators, but we did not specifically ask for a report on green olive intake. However, participants did not report consuming any olives when reporting their

daily nutritional habits. In addition, the background intake of Oleuropein in healthy populations is much lower than the dose we used for the intervention, and any minor effect would be corrected by the randomization and cross-over design between placebo and OLE intervention.

Hypothesis. PDH activity as a proxy for mitochondrial Ca²⁺ uptake, and do not measure mitochondrial Ca²⁺ contents (which admittedly is difficult). yet, you state that the purpose is to test if OLE ".is triggering mitochondrial Ca²⁺ uptake and PDH activation during MICE. You only measure PDF activity and not mitochondrial Ca²⁺ contents, thus this is not valid to state both.

We agree that this sentence is confusing and could give the impression that we directly measured mitochondrial calcium uptake. Therefore, we have removed or reported in conditional form all mentions of mitochondrial calcium uptake from the hypothesis statement in the Discussion section.

Biopsies. The central conclusions about OLEs modulation of inflammation and enhancement of mitochondrial pathways rely on bulk RNA-seq and proteomics immediately post-exercise and at 24 h (Figures 5, 6, and 7). Pre and Post biopsies were collected through the same incision, with multiple samples from same puncture site. By using the same incision for both Pre and Post muscle biopsies could introduce local injury/inflammation that confounds signals attributed to exercise or OLE (Figs 5-7). Thus, repeated needle biopsies cause disturbances in myocellular signalling pathways (doi:10.14814/phy2.286). I'm concerned if the acute immune pathway enrichments and OLE effects are affected by biopsy-induced tissue injury and thereby the interpretation of acute inflammatory and mitochondrial pathway changes. Please comment on this possibility and justify.

We acknowledge that post-intervention measurements of biopsy-related parameters—particularly those involving immune responses—may have been partially influenced by tissue injury caused by the pre-intervention biopsy. However, evidence suggests that such effects are limited within the time frame relevant to our study (30–60 min)(doi:10.1152/jappphysiol.00091.2011). In addition, while this could potentially affect the response to exercise in our design, the effects of repeated biopsy will not affect the effects of OLE as OLE and placebo interventions were controlled via a cross-over design where half of the participants received OLE first and the other half received placebo first, with the same biopsy schedule during each visit.

Mitochondrial effects. The proteomic and targeted analyses revealed a downregulation of mitochondrial-related pathways immediately after MICE while only proteomics indicated a downregulation after SIE. Decreases were observed for Complex III, IV, and V subunits after MICE, while no changes after SIE. Further, the mitochondrial pathway downregulation was reasoned from proteomic enrichment. As the protein analysis was assessing only one subunit per OXPHOS complex and without normalization to mitochondrial content markers or broader mitochondrial protein panels, limits the generalization to pathway-level downregulation and the assertion that only proteomics detected changes after sprint interval exercise. Studies on effects of exercise on OXPHOS subunits show a non-stoichiometric change (mitochondrial contents / analysed) in response to acute exercise, and most training-induced changes in mitochondrial functional groups become non-significant when normalized to mitochondrial content markers as e.g. CS. As the data presented are a narrow set of proteins estimates of mitochondrial abundance, the observed decreases and the lack of changes may not represent widespread pathway behaviour. Please justify the claims.

We totally agree with the reviewer that manual investigation of one subunit per complex may not be sufficient to indicate a complex specific alteration and that SIE may not be as affected than MICE. To make this clear the following sentence from the *Mitochondrial responses of OLE ingestion combined with MICE and SIE* section was added in the manuscript: “As the investigation of one subunit per complex may not comprehensively represent the mitochondrial related response, we also annotated the proteomic data with the MitoCarta database and performed an enrichment analysis.”

The enrichment analyses indeed confirms that the mitochondrial changes do not seem to be complex specific but rather reflect general mitochondrial alterations. We modified the results as well as the discussion section accordingly.

PDH activity. PDH activity was increased following MICE only when combined with OLE, while OLE did not further increase PDH activity in response to SIE. The PDH activity in the homogenized biopsy material was estimated using a standard assay kit. The homogenisation buffer contained 5 mM EGTA and 5 mM dichloroacetate (DCA), which buffer Ca^{2+} (EGTA) and is known to inhibit PDK (DCA). Thus, I fear that the assumed Ca^{2+} dependent activation of PDH, i.e. preventing the Ca^{2+} dependent dephosphorylation of PDH (e.g. doi: 10.1042/bj2180235), is confounded and interpretation of PDH activation as a proxy for mitochondrial Ca^{2+} uptake is not valid. Further, DCA is known to inhibit PDK thereby activating PDH in a non- Ca^{2+} depending way, (e.g. doi.org/10.1210/jc.2005-0123). Together, the observed increase in PDF activity can be attributed to (or possibly are) homogenising buffer, weakening the mechanistic link central to the claim. Together, the assays need to be performed in Ca^{2+} controlled and DCA-free conditions to verify the assay validity and hence mechanistic claim.

PDH assay was performed in controlled conditions, following standard procedure (e.g.: Costantin-Teodosiu et al., *Analytical Biochemistry*, 1991; 198:347-351; A sensitive radioisotopic assay of pyruvate dehydrogenase complex in human muscle tissue). All measurements were done in parallel in a randomized order and all samples were continuously maintained on ice to maximize preservation of the intact PDH phosphorylation at the time of biopsy collection. The guiding principle for adding these chelators/inhibitors was precisely to further prevent any ex vivo modification of the PDH phosphorylation state after biochemical isolation. NaF was added in the homogenization buffer as a general phosphatase inhibitor to prevent PDH dephosphorylation. EGTA and dichloroacetate (DCA) are commonly included in PDH activity measurement protocols during tissue or cell homogenization to inhibit PDH dephosphorylation by the Ca^{2+} -sensitive PDP and phosphorylation by PDK, respectively (see for example e.g.: Kerr et al. (2011), *Assays of Pyruvate Dehydrogenase Complex and Pyruvate Carboxylase Activity*, in *Mitochondrial Disorders: Biochemical and Molecular Analysis*, P.D.L.-J.C. Wong, Editor. 2012, Humana Press: Totowa, NJ. p. 93-119). Using this standard assay thus ensures that PDH activity is measured in a situation where its in vivo phosphorylation state is preserved. We have added an explanation about this rationale in the methods.

Mitochondrial Ca^{2+} buffering. Lines 629-631 and throughout the manuscript its stated that mitochondrial Ca^{2+} buffering can modulate cell metabolism. I understand its debated and mitochondrial Ca^{2+} uptake activates metabolic activity, however, as there are no apparent Ca^{2+} buffers in the mitochondria the Ca^{2+} buffering capacity is relatively limited. Please discuss the issue of mitochondrial buffering and quantitative importance for cellular Ca^{2+} regulation.

We acknowledge that the contribution of mitochondria to cellular Ca^{2+} buffering remains a matter of debate (DOI: 10.1085/jgp.202213167). As Ca^{2+} buffering is not the main interest of the present study we decided that removing the mention to Ca^{2+} buffer would clarify the discussion and limit any confusion.

Fatigue index. Lines 435-440. The fatigue index across six sprints and the first sprint, showing a significant reduction with the supplement only in the first sprint. The improvement in early-stage fatigue resistance is inferred from a lower fatigue index in the first 30-second sprint with olive leaf extract (Fig 2E), alongside the progression of fatigue index across sprints (Fig 2D) and the comparison of first sprint mean power (Fig 2C). The analysis does not rule out that the lower fatigue index in the first sprint reflects differences in initial power or acceleration rather than true resistance to fatigue. The fatigue index depends on the change between the first and last 5 seconds of the sprint (Fig 2E), and differences in early power profile will affect this index. Were there any differences in initial power between trials?

It is correct that the fatigue index can be affected by power at the start of each sprint. We observed no differences on power produced during the first 5 seconds of the 1st sprint between PLA and OLE (11.0 ± 1.05 vs 10.4 ± 0.98 W/kg, $p=0.15$).

This result has been added in the text which exclude a confounding effect of initial power output.

Heart rate (p. 15). The stated reduction in mean heart rate during moderate cycling does not meet statistical significance, i.e. p-value 0.0547 (Fig 2A). However, there is definitely a trend-level which likely may be constrained by sample size. Please rewrite and modify the claim to reflect this.

We thank the reviewer for notifying this. We modified the result section and the discussion according to the comment.

Inflammatory signalling. The overall conclusion that OLE enhances acute inflammatory signalling after MICE and attenuates it after SIE at 24 h is based on pathway enrichment analysis of bulk muscle RNA-seq data. Measures of "HALLMARK pathways" (Fig 6A-B) exhibit consistent transcript changes, and proteomics data (Figures 5C-D) confirm that exercise induces immune and inflammatory responses. This is carefully generated. Still, inflammatory modulation is concluded from RNA-seq pathways without direct protein-level or cellular validation. I'll suggest that the authors clearly state and discuss that the mRNA changes may not reflect actual cytokine abundance or signalling activity, as well as the proteomics analysis does not specifically assess OLE-related inflammatory effects.

We thank the reviewer for this relevant comment. We now specify in the section discussing inflammatory modulation that the observed response represents a "transcriptional" response. In addition, we have added a brief paragraph to the limitations to acknowledge that changes in mRNA of inflammatory pathways may not always translate into the actual abundance of cytokines or pathway

activation.

Also, the acute comparison between modalities is confounded by sampling times for the Post biopsy sampling time relative to exercise start and OLE administration. This may affect or bias the detection of immediate effects after SIE. The observed lack of OLE effect after SIE could reflect insufficient time for transcript, which is observed in MICE where biopsy is taken noticeably later leaving changes to manifest. Please discuss that in context to the overall conclusion that OLE enhances acute inflammatory signalling after MICE and attenuates it after SIE.

We agree with the reviewer that the different intervals between the start of exercise and the Post biopsy for MICE and SIE may confound the effect of OLE on the Post transcriptional response, particularly after SIE. We already address this in the second paragraph of the discussion as a potential explanation for the lower number of regulated genes observed after SIE. We have now added a sentence in the paragraph discussing the inflammatory response to highlight this point in that context as well.

Assessments. You very precisely assessed both neuromuscular parameters and torque and EMG during knee extension. Its not clear by the reader why and relevance of these measures for the overall aim of test the hypothesis that " .. combining OLE to SIE or MICE may influence muscle bioenergetics .. . I may have missed it, but please explain the purpose using these estimates and possible interpretations of the results. Also, lines 225-229, please explain briefly the assumption that PS10/100 is an index of intracellular Ca²⁺ handling. Do you mean release?

We thank the reviewer for highlighting this point. Since some metabolic products are known to influence fatigability, and OLE is expected to enhance mitochondrial bioenergetic during exercise, we explored whether OLE might affect exercise-induced neuromuscular fatigue. Accordingly, we have added this as a secondary aim at the end of the Introduction.

PS10/100 is an index used to assess prolonged low-frequency force depression, previously referred to 'low frequency fatigue' (doi.org/10.1113/jphysiol.1977.sp012072 ; doi.org/10.1152/japplphysiol.01051.2007), a form of fatigue characterized by a disproportionate loss of force in response to low-frequency electrical stimulation. Prolonged low frequency force depression is closely linked to impairments in Ca²⁺ handling, which can originate from: (1) reduced Ca²⁺ release from the sarcoplasmic reticulum (SR), (2) impaired sensitivity of the contractile filaments to Ca²⁺, and/or (3) reduced Ca²⁺ reuptake into the SR or other buffering systems. Previous works on mouse intact muscle fibres indicate that prolonged low frequency force depression mainly originates from impaired SR Ca²⁺ release (DOI: [10.1152/jappl.1993.75.1.382](https://doi.org/10.1152/jappl.1993.75.1.382)). We clarified this point in the method.

Title. Please be more specific in title on "mitochondrial response" its unclear to the reader what meant. I understand that it's a relatively broad range of mitochondrial affects you are determining, but consider using more specific term(s), i.e. mitochondrial transcriptional response and respiratory complex protein levels.

We modified the title as "Oleuropein-based olive leaf extract enhances muscle mitochondrial bioenergetics response to moderate –but not maximal- intensity exercise in humans "

Minor

Lines 68-71. Off note, the leaky RyR1 was observed only in untrained subjects/unaccustomed exercise. While trained still have "beneficial metabolic adaptations" with no apparent RyR1 leak. Please state clearly that training adaptations is only partly explained by the Ca²⁺ Leak/release effects.

We specified that the leak of RyR1 following SIT has been observed in healthy active men and that this may be "... responsible for some of the beneficial metabolic adaptations..."

Line 79. Therefore, we wondered State clearly if this is a hypothesis. Likewise, line 101, ... we reasoned I believe you hypothesize.

We reworded these two sentences to clearly state our hypotheses.

Introduction. Lines 61.62. The justification of SIE training as being more "time efficient" is a common accepted assertion. A quick calculation reveals that your MICE protocol last just more than 1 hr and the SIE, all in all 30 min. It would be more relevant to state the superior effects on circulation and health parameters with HIIT/SIE compared to MICE, due to also activation of fast twitch fibres.

We do not fully agree with the statement that SIT is superior to MICT for improving health parameters. In healthy populations, a recent review indicates that while SIT may confer slightly greater benefits during the initial weeks of training, longer-term interventions show that MICT matches—or even outperforms—SIT (<https://doi.org/10.1007/s40279-024-02120-2>). That said, it is true that SIT recruits a greater proportion of type II fibers than MICT due to the higher exercise intensity. However, it remains unclear whether this enhanced recruitment translates into additional muscle adaptations as compared with MICT.

Based on this argumentation, we believe that our introduction is consistent with the existing literature and does not need to assert a clear superiority of SIT over MICT.

Figure 9. The increased PDH activity post SIE in PLA (Fig 9B) seems driven by one person. Are there any effects omitting the subject?

If we remove this participant, the p-value = 0.055. In our opinion, this does not fundamentally change the interpretation of the data. We kept all the participants as we have no specific reason to exclude this one in particular.

Line 595. "Since OLE activates bioenergetics by stimulating PDH activity via increased mitochondrial Ca²⁺ uptake during contraction, Please add reference.

The reference "Gherardi et al., 2024" corresponding to this statement has been added.

Line 120. Did you estimate VO₂ during the incremental cycling protocol. i.e. VO₂max? and how was MAP defined, i.e. last

Gaz exchanges were not measured during the incremental protocol. Therefore, VO₂max could not be measured. As explained in the "Familiarization" section, the MAP was defined as the last power

reached during the incremental test (no average).

Line 435. Power decrease, give P-value.

The p-value has been added.

Line 437. How is "anaerobic performance" defined.

Based on the work of Smith & Hill, 1991, anaerobic performance can be defined as the power output generated during an effort in which a large proportion of ATP resynthesis is supplied by anaerobic metabolic pathways. This has been clarified in the text.

Referee #2:

I have been reading the paper by Lanfranchi and colleagues, where they are looking into different training modalities effect on mitochondrial bioenergetics, mitochondrial calcium uptake and pyruvate dehydrogenase activation. Furthermore they also looked into the compound oleuropeins effect on these parameters mentioned before. The study was conducted as a double-blinded cross-over study in healthy males. Muscle biopsies were obtained before, immediately after and 24 hours after an acute exercise bout (moderate intensity: 1 h at 50% maximal aerobic power or sprint interval exercise 6 × 30 s all-out sprints with 4 min recovery). I have some comments to the manuscript.

We thank the Reviewer for the appreciation of our manuscript. Please find point-by-point responses below.

Comments:

Line 73-75: The study by Zanou et al. was conducted in C2C12 cells as I remember, maybe add this information.

Our previous study was conducted both in human and C2C12 myotubes as an in vitro model of exercise. We now specify this in the text.

Line 77: What clinical populations are unable to perform intense exercise, please clarify that and add a reference.

We acknowledge that an increasing number of publications report that high-intensity exercise is safe for most individuals. However, these protocols are often adapted and differ than the ones used in healthy population. Overall, MICT is easier to implement in patients as compared to HIIT/SIT even though HIIT/SIT paradigm is doable in patients and show benefits.

We revised this paragraph to emphasize the broad benefits of enhancing metabolic adaptations through exercise, applicable to both the general population and individuals with clinical conditions.

Line 110: The introduction are focusing a lot of health burden and chronic diseases, why then investigate it in young healthy men and why only men?

By revising the paragraph that previously addressed the limited capacity of clinical populations to perform high-intensity exercise, we believe the introduction now places less emphasis on clinical considerations.

The use of OLE for its effect on skeletal muscle metabolism in combination with exercise had never been investigated in human before. Therefore, as a first step to validate the potential benefit of this intervention, we decided to perform this study in healthy volunteers before switching towards clinical population, which will be the focus of an upcoming study in our laboratory. The inclusion of only men was due to difficulties to recruit women and the extended time that would have been required to enrol a sufficient number of female participants to meaningfully consider sex as a biological variable.

Line 112: What is the meaning of good health and stable eating habits, please clarify that.

This statement was indeed unclear and has been removed. The sentence now reads: "Prior to starting the experimentation, all the participants gave written informed consent and completed a health questionnaire to exclude individuals who might be at risk during the exercise protocol or the biopsy procedure."

Line 114: Should it not be "performed at..."

This has been corrected.

Line 128: Was the taste of the product and the placebo the same?

Yes, the encapsulation prevented from any specific taste to be detected by the participants, and the remainder of the capsule was identical to the placebo, making the two indistinguishable. This is now specified in the text.

Line 135: Why was this timepoint (30 min after ingestion) picked?

Oleuropein aglycone has been shown to peak in plasma between 30 and 90 minutes after olive leaf extract ingestion (Pinckaers et al., 2025). Because the biopsy was performed approximately 30 minutes after ingestion, the exercise bout could begin around 45–50 minutes post-ingestion and therefore occurred during the period when plasma oleuropein levels were elevated.

Line 161-164: Has the Authors thought about if there is any issues with their analysis and the sampling of muscle biopsies? Could this in any way affect their outcome? Please add this in the revised manuscript.

We have added a short paragraph in the limitations (end of discussion section) to acknowledge that some of the Post biopsy-related measurements may have been partly confounded by the repeated biopsy sampling.

Line 195: What was the Authors experience with compensatory movements?

The position of the participants was standardized by using a belt across the abdomen and two cross-shoulder harnesses to prevent any undesired movements. This was especially needed for the quantification of the maximal voluntary contraction force. Although we cannot totally exclude that other muscles contributed to the force produced during the maximal knee extensions (e.g. abdominal muscles), the standardized position certainly limited the variation of this contribution, i.e. repetitions from the same participant can be reliably compared.

Line 205: Was the two exercise protocols matched for energy used?

No, the two exercise protocols were not matched for total work. The goal was to assess the effect of both exercise modalities as they are usually performed. In the present study the total amount of work produced during MICE was around 7 times higher than for the SIE protocol.

Line 264: How long time after the muscle biopsies was taken was the analysis done? How long was the samples stored in the freezer?

All analyses of the muscle biopsies were conducted between six months and one year after the completion of data collection. This information has been added at the end of the “Muscle microbiopsies” section.

Line 418: Please clarify what several issues were?

The issues are listed following this sentence. We modified the text to clarify it as follows: “The variation in the *n* values from the initial inclusion is due to the following issues. For performance data of the SIE group the recording of one participant was not correctly saved. After the first microbiopsy, one participant from the MICE group decided to stop the biopsy procedures but he was still included for the other parameters. One participant from the MICE group was removed from the western blot analyses due to poor quality of the sample and incoherent protein levels. For the PDH activity, measures of two samples were out of the standard curve and were therefore excluded.”

Line 508: Very interesting, where the biopsies in the present study taken at the same time as the study by Layec, 2018?

In their study, Layec performed the biopsies within 30s following termination of exercise while in our case the biopsy was performed around 3-4min after the termination of exercise. We believe that these results can be compared as it is unlikely that protein abundance significantly changed between 30s and 4min of recovery post exercise.

Line 524-525: What is meant by slightly lower?

This sentence has been modified to limit subjective wording. It is now stated that “...the response of these pathways 24 h post MICE was similar or even decreased in participants consuming OLE.”

Line 567-568: It is surprising for me that the protein levels changes that fast, could the Authors please elaborate on that?

We thank the reviewer for highlighting this point. We were also surprised to see a consistent decrease in mitochondrial proteins immediately after exercise. This may align with recent evidence suggesting that acute exercise may initiate mitophagy and mitochondrial ejection from muscle fibers (<https://doi.org/10.1111/apha.14203>). However, this hypothesis remains very speculative and further studies should be specifically designed to tackle this question.

Line 594: As far as I can see calcium uptake was not measured in the present study, why not?

We agree that a direct measurement of mitochondrial calcium uptake would have more precisely answered our research question. However, mitochondrial calcium uptake measurement in human muscle fibres in this context is very challenging and was therefore not performed in the present study.

Dear Dr Place,

Re: JP-RP-2026-290316R1 "Oleuropein-based olive leaf extract enhances muscle mitochondrial bioenergetics response to moderate -but not maximal- intensity exercise in humans" by Clément Lanfranchi, Alba Moreno-Asso, Astrid MH Horstman, Sara Mistro, Eugenia Migliavacca, Ornella Cominetti, Jens Stolte, Sylviane Métairon, Aurélie Hermant, Ane Laura Pedersen, Loïc Dayon, Umberto De Marchi, Jerome Feije, Nadège Zanou, and Nicolas Place

Thank you for submitting your manuscript to The Journal of Physiology. It has been assessed by a Reviewing Editor and by 2 expert referees and we are pleased to tell you that it is acceptable for publication following satisfactory revision.

REVISION CHECKLIST:

Please upload two versions of your manuscript text: one with all relevant changes highlighted and one clean version with no changes tracked. The manuscript file should include all tables and figure legends, but each figure/graph should be uploaded as separate, high-resolution files. The journal is now integrated with Wiley's Image Checking service. For further details,

see: <https://www.wiley.com/en-us/network/publishing/research-publishing/trending-stories/upholding-image-integrity-wileys-image-screening-service>

We look forward to receiving your revised submission.

Yours sincerely,

Paul Greenhaff
Senior Editor
The Journal of Physiology

EDITOR COMMENTS

Reviewing Editor:

Comments to the Author:

Authors have done a very nice job of addressing the comments and suggestions from the two reviewers. However, reviewer 1 has one remaining minor point, which require authors attention.

Senior Editor:

Comments to the Author:

Thank you for the revised manuscript and rebuttal document which have been considered by the same reviewers that considered the original submission. All involved were positive about the response of the authors in the rebuttal document and the revised manuscript that has been improved. Reviewer 1 has a further minor comment concerning the revisions made by the authors (that the Reviewing Editor believes is fair), which the authors are requested to address. Thank you.

REFEREE COMMENTS

Referee #1:

Thank you for responding thoroughly to my comments and adequately revised the text. I only have one remaining issue regarding my comment that the increased PDH activity post SIE in PLA (Fig 9B) seems driven by one person. The comment was not a suggestion to remove the data point, rather, a more nuanced view on the data, with the still limited number of subjects included in the study. Although the P-value in this respect is not key interpreting the data, it still of interest for the reader to know if its driven by few/one data point. Thus, Ill suggest mentioning that when disregarding the value P equals 0.055, or similar text.

Referee #2:

I would like to thank the Authors for revision of the manuscript. They have answered all my concerns in their letter.

END OF COMMENTS

EDITOR COMMENTS

Reviewing Editor:

Comments to the Author:

Authors have done a very nice job of addressing the comments and suggestions from the two reviewers. However, reviewer 1 has one remaining minor point, which require authors attention.

Senior Editor:

Comments to the Author:

Thank you for the revised manuscript and rebuttal document which have been considered by the same reviewers that considered the original submission. All involved were positive about the response of the authors in the rebuttal document and the revised manuscript that has been improved. Reviewer 1 has a further minor comment concerning the revisions made by the authors (that the Reviewing Editor believes is fair), which the authors are requested to address. Thank you.

We thank the Reviewing Editor and the Senior Editor for their positive assessment of our work. Please find below the answer to the comment of Referee 1.

We also added the dataset identifiers under the “Data availability” section for both proteomics and transcriptomics.

REFEREE COMMENTS

Referee #1:

Thank you for responding thoroughly to my comments and adequately revised the text. I only have one remaining issue regarding my comment that the increased PDH activity post SIE in PLA (Fig 9B) seems driven by one person. The comment was not a suggestion to remove the data point, rather, a more nuanced view on the data, with the still limited number of subjects included in the study. Although the P-value in this respect is not key interpreting the data, it still of interest for the reader to know if its driven by few/one data point. Thus, Ill suggest mentioning that when disregarding the value P equals 0.055, or similar text.

Thanks for the clarification. We now disclose in the text that one participant had an important increase in the post measurement and that removing this participant from the statistical analysis would give a p-value = 0.055.

We also modified the Panel 9B. We removed the stars indicating significance and added the main effect and post-hoc comparison represented in figure 3.

Dear Professor Place,

Re: JP-RP-2026-290316R2 "Oleuropein-based olive leaf extract enhances muscle mitochondrial bioenergetics response to moderate -but not maximal- intensity exercise in humans" by Clément Lanfranchi, Alba Moreno-Asso, Astrid MH Horstman, Sara Mistro, Eugenia Migliavacca, Ornella Cominetti, Jens Stolte, Sylviane Métairon, Aurélie Hermant, Ane Laura Fineid Pedersen, Loïc Dayon, Umberto De Marchi, Jerome N Feige, Nadège Zanou, and Nicolas Place

We are pleased to tell you that your paper has been accepted for publication in The Journal of Physiology.

IMPORTANT

(1) You must start the Methods section with a paragraph headed Ethical Approval. If experiments were conducted on humans, confirmation that informed consent was obtained, preferably in writing, that the studies conformed to the standards set by the latest revision of the Declaration of Helsinki and that the procedures were approved by a properly constituted ethics committee, which should be named, must be included in the article file. If the research study was registered (clause 35 of the Declaration of Helsinki), the registration database should be indicated, otherwise the lack of registration should be noted as an exception (e.g. The study conformed to the standards set by the Declaration of Helsinki, except for registration in a database). For further information see: <https://physoc.onlinelibrary.wiley.com/hub/human-experiments>.

(2) We have a note from Astrid Horstman that her current affiliation on the title page needs to be changed: 'I confirm the submitted manuscript is the version that I approved. However, I would like to change my current affiliation from "Human Performance Lab, Schulthess Clinic, Zurich, Switzerland" to "Faculty of Health, Nutrition and Sport, The Hague University of Applied Sciences, The Hague, The Netherlands".'

Please can you confirm whether or not your study was a registered clinical trial, and email us an updated article (Word) file that (a) includes clinical trial details (or a statement to say that the study was not registered) and (b) includes an updated title page with Astrid's new current affiliation.

You can email the file to us at: jp@physoc.org

Yours sincerely,

Paul Greenhaff
Senior Editor
The Journal of Physiology

IMPORTANT POINTS TO NOTE FOLLOWING ACCEPTANCE OF YOUR PAPER:

- **IMPORTANT NOTICE ABOUT OPEN ACCESS:** To assist authors whose funding agencies mandate immediate public access to published research findings, The Journal of Physiology allows authors to pay an Open Access (OA) fee to have their papers made freely available immediately on publication.

The Corresponding Author will receive an email from Wiley with details on how to register or log in to Wiley Authors where you will be able to place an order.

- You can check if your funder or institution has a Wiley Open Access Account here:

<https://authors.wiley.com/author-resources/Journal-Authors/open-access/author-compliance-tool.html>

- You can help your research get the attention it deserves! Check out Wiley's free Promotion Guide for best-practice recommendations for promoting your work at: www.wileyauthors.com/eeo/guide. You can learn more about Wiley Editing Services which offers professional video, design, and writing services to create shareable video abstracts, infographics, conference posters, lay summaries, and research news stories for your research at: www.wileyauthors.com/eeo/promotion.

- If you would like to receive our 'Research Roundup', a monthly newsletter highlighting the cutting-edge research published in The Physiological Society's family of journals (The Journal of Physiology, Experimental Physiology, Physiological Reports, The Journal of Nutritional Physiology and The Journal of Precision Medicine: Health and Disease), please click this link, fill in your name and email address and select 'Research Roundup':

<https://www.physoc.org/journals-and-media/membernews>

EDITOR COMMENTS

Reviewing Editor:

Authors have adequately handled this final issue.

Senior Editor:

Thank you for the minor revision. The manuscript is acceptable for publication. Congratulations and thank you for supporting The Journal of Physiology.